# SALMAN: Stability Analysis of Language Models Through the Maps Between Graph-based Manifolds

## Abstract

Recent strides in Pretrained Transformer-based language models have propelled state-of-the-art performance in numerous NLP tasks. Yet, as these models grow in size and deployment, their robustness under input perturbations becomes an increasingly urgent question. Existing robustness methods often diverge between small-parameter and large-scale models (LLMs), and they typically rely on labor-intensive, sample-specific adversarial designs. In this paper, we propose a unified, local (sample-level) robustness framework (SALMAN) that evaluates model stability without modifying internal parameters or resorting to complex perturbation heuristics. Central to our approach is a novel Distance Mapping Distortion (DMD) measure, which ranks each sample's susceptibility by comparing input-to-output distance mappings in a near-linear complexity manner. By demonstrating significant gains in attack efficiency and robust training, we position our framework as a practical, model-agnostic tool for advancing the reliability of transformer-based NLP systems.

## 1 Introduction

Recently, breakthroughs in pretrained Transformer-based language models have revolutionized the field of Natural Language Processing (NLP). These models have enhanced performance across a wide range of downstream NLP tasks, including text classification Sun et al. (2019), summarization El-Kassas et al. (2021), chatbot Achiam et al. (2023), and complex reasoning Wei et al. (2022). Given the widespread adoption of language models, it is crucial to evaluate their robustness. The robustness problems in the NLP community mostly focus on exploring the language model behavior when the inputs are modified.

Ebrahimi et al. (2017) design character-level and word-level perturbations as adversarial examples to attack NLP models. Jia & Liang (2017) explore the method to mislead the language model's output in the Q&A task by adding random sentences. Later, research such as that by Jin et al. (2020) and Li et al. (2020) focused on designing adversarial samples that better preserve the original semantics. Subsequently, some work began to analyze the robustness of language models in continuous space and improve the generalization ability of NLP models and defense against word substitution attacks through adversarial training in continuous space Zhu et al.; Li et al. (2021; 2023). Recently, the growth of model parameter size and training data for language models has demonstrated that Large Language Models (LLMs) exhibit increased robustness to trivial disturbances and handle common disruptions more effectively Achiam et al. (2023); Zou et al. (2023). As a result, recent studies have devised Jailbreak prompts specifically designed to attack LLMs, thereby evaluating and testing their robustness Wang et al. (2023); Zhu et al. (2023b).

While significant progress has been made in the robust evaluation of pre-trained language models, current robustness analyses still face several key challenges. First, the methods for evaluating robustness in large language models (e.g., LLaMA-series) and those in smaller models (e.g., BERT, BART) differ substantially. In smaller models, word-level or token-level adversarial attacks often suffice to expose vulnerabilities Li et al. (2018; 2020); Garg & Ramakrishnan (2020). However, large language models can often interpret or adapt to these simple perturbations, necessitating more carefully designed prompt-based strategies for effective robustness testing Zou et al. (2023); Paulus et al. (2024). As a result, a unified robustness evaluation framework applicable to both large and

small models is currently lacking. Second, irrespective of a model's parameter size, designing adversarial inputs or prompts remains a time-consuming and labor-intensive process, particularly for large-scale NLP datasets. These challenges highlight the need for automated, scalable, and more universal approaches to evaluate the robustness of diverse language models.

In this work, we propose a sample-centric robustness framework that addresses both challenges by: 1) Unifying Robustness Evaluation for All Model Scales. Specifically, our method computes a local (per-sample) robustness measure, applicable to both smaller models (e.g., DistilBERT) and massive LLMs (e.g., GPT-2, Llama) *without* requiring changes to their internal parameters or specialized adversarial training. 2) Minimizing Labor-Intensive Perturbation Design. Instead of heavily relying on constructing adversarial prompts, we quantify each input's inherent vulnerability via a lightweight, near-linear complexity approach. This ranking guides adversarial attacks or fine-tuning decisions, drastically reducing manual effort compared to purely sample-by-sample adversarial generation.

At the core of our method is a novel, per-sample distance mapping distortion (DMD) metric that compares distances in the input representation space against distances in the output representation space. To facilitate these distance calculations efficiently, we first build a *near-linear complexity* Probabilistic Graphical Model (PGM) that captures the manifold structure of the data, preserving both local geometry and global structural properties without resorting to dense or iterative global optimizations. By assessing each instance individually, we gain a fine-grained view of where and how a model fails to preserve distances across its representations—leading directly to broader insights about the system's behavior as a whole. Such per-sample analyses, in turn, form the building blocks of understanding overall stability Zhang et al. (2019a). Rather than relying on aggregate statistics alone, examining each sample's distortion enables us to pinpoint particular modes of fragility. We show how this ranking can: 1) Streamline NLP Attacks: Targeting non-robust samples first yields more efficient and more effective adversarial attacks (Section 4.2). 2) Improve LLM robustness through Fine-Tuning: Up-weighting non-robust data during fine-tuning preserves or even improves generalization and yields internal representations closer to the pre-trained checkpoint (Section 4.3). Furthermore, the same method applies to both smaller-scale models and large-scale LLMs, offering a unified pathway for robustness analysis across diverse parameter regimes.

Overall, our contributions are:

- A unified robustness measure (SALMAN) that can be computed in nearly-linear time for language models of varied sizes (from smaller transformers to LLMs), without requiring specialized tasks, perturbed data, or parameter modifications.

- To our best knowledge, SALMAN is the first local (sample-level) robustness measurement specifically tailored from small to large language models , enabling fine-grained analysis of how individual inputs withstand minor or adversarial perturbations.

- Empirical demonstrations across both small (BERT, DistilBERT) and large models (GPT-2, Llama) showing how this sample-level perspective leads to (i) more efficient and higher success-rate adversarial attacks, and (ii) improved robust fine-tuning outcomes.

## 2 BACKGROUND

### 2.1 ROBUSTNESS IN NLP

The robustness of language models remains a pivotal area of research within the NLP community. Several studies have explored the vulnerability of these models to modifications in the input text, ranging from typos to word replacements Li et al. (2020); Jin et al. (2020); Sun et al. (2020). Wang et al. (2021) further developed a multi-task benchmark to evaluate language model robustness. In the realm of model probing, Tenney (2019) and Hewitt & Manning (2019) examined how syntactic and semantic features are represented across different layers of BERT-like models. Voita et al. (2019) and Abnar et al. (2019) employed similarity-based analysis methods to study the evolution of representations in deep neural networks. Zhou & Srikumar (2021) and Neerudu et al. (2023) performed a comprehensive analysis of how finetuning affects the representations in the language model using a combination of probing and analytical techniques. With the increase in model parameters, LLMs can distinguish between minor textual variations, underlining the need to explore their robustness to input perturbations. Recent studies have focused on the impact of input prompts on LLM robustness Shayegani et al. (2023). Wang et al. assessed the robustness of ChatGPT against adversarial

and out-of-distribution samples. Zou et al. (2023) enhanced the efficiency of jailbreak attacks by generating adversarial suffixes. DecodingTrust examined the robustness of LLMs using standard datasets like AdvGLUE and AdvGLUE++ Wang et al. (2023). PromptRobust investigated the robustness of LLMs from the perspective of prompts, demonstrating that subtle changes in instructions can lead to significant performance variations Zhu et al. (2023b).

## 2.2 PROBABILISTIC GRAPHICAL MODELS

Probabilistic Graphical Models (PGMs) represent conditional dependencies among variables in a graph, enabling interpretability and efficient inference Koller (2009). Here, each sample (e.g., a Transformer embedding) becomes a node, with edges capturing local/global interactions that approximate the data manifold Vu & Thai (2020); Feng (2021); Rubin-Delanchy (2020). Tightly connected subgraphs indicate higher intrinsic similarity, while loosely connected regions suggest divergence. Recently, SAGMAN Cheng et al. (2024) extends these PGM-based ideas to GNNs by incorporating dimension reduction for domain-specific manifold structures.

## 3 THEORETICAL FOUNDATIONS OF SALMAN

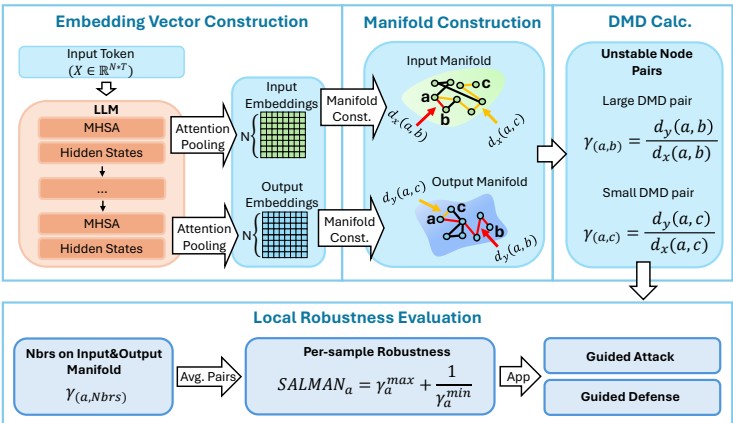

Figure 1: The overview of SALMAN Method.

In this section, we detail our overall pipeline (Figure 1), followed by the mathematical underpinnings of (1) Embedding Construction (Sec. 3.1), (2) Manifold Construction (Sec. 3.2), (3) Distance Mapping Distortion (Sec. 3.3), and (4) Algorithm Complexity Analysis (Sec. 3.4).

## 3.1 EMBEDDING VECTOR CONSTRUCTION

A key challenge in analyzing transformer-based language models arises from the *discrete* nature of token embeddings, which may not form well-behaved manifolds in the topological sense (Robinson et al., 2024). In particular, the geometry of the token space is heavily fragmented: a small textual perturbation (e.g., replacing a token with a synonym) can induce a disproportionately large jump in token-level embedding indices. As a result, continuous manifold-based analyses, which rely on smooth neighborhoods and gradual changes, become intractable when applied directly to discrete tokens. Moreover, transformers often exhibit *stochastic decoding* (via temperature sampling, beam search, etc.) Li et al. (2024), meaning identical input text can produce slightly different token outputs. Hence, relying purely on discrete token sequences introduces variability that disrupts stable manifold construction.

**Attention Based Embedding Representation.** To address these issues, we aggregate each sample's token embeddings into a *single* continuous vector, thereby avoiding the discontinuities of the raw token space. Formally, given a natural language dataset of $N$ samples, each data sample is tokenized into a sequence of embeddings $\{\mathbf{x}_1, \ldots, \mathbf{x}_N\}$. We then pass each $\mathbf{x}_i$ through a Transformer-based pre-trained language model to obtain its Multi-Head Self-Attention (MHSA) *outputs*, which we denote as $A_i = \text{MHSA}(\mathbf{x}_i) \in \mathbb{R}^{H \times T_i \times d_{\text{model}}}$. $H$ is the number of attention heads, $T_i$ is length of $\mathbf{x}_i$, and $d_{\text{model}}$ is the hidden dimension. Then, we average these per-head outputs along

the head dimension: $\overline{A}_i = \frac{1}{H} \sum_{h=1}^{H} A_{i,h} \in \mathbb{R}^{T_i \times d_{\text{model}}}$. Next, we *compute an attention-based weighting*. Softmax is applied to attention matrices $\overline{A}_i$ over the $T_i$ tokens to obtain $\boldsymbol{\alpha}_i \in \mathbb{R}^{T_i}$. Finally, we compute a single pooled vector $\mathbf{v}_i \in \mathbb{R}^{d_{\text{model}}}$ for each sequence by a weighted sum of the token embeddings: $\mathbf{v}_i = \sum_{t=1}^{T_i} \boldsymbol{\alpha}_{i,t} \overline{A}_{i,t}$. Collecting all $\mathbf{v}_i$ for $i = 1, \ldots, N$ then yields a new embedding matrix $\{\mathbf{v}_1, \ldots, \mathbf{v}_N\}$. We denote embedding matrix taken from the first layer of the language model as $\mathbf{z}_X$ and embedding matrix taken from the last layer of the language model as $\mathbf{z}_Y$.

**Deterministic Hidden-State Embeddings.** Though transformers can produce stochastic token outputs, the internal hidden states remain largely deterministic *if we freeze the model parameters and inference procedure*. For instance, by disabling dropout layers and using a fixed random seed, we empirically found that $\mathbf{z}_Y$ becomes stable irrespective of minor token-level variations. Specifically, we measured multiple $\mathbf{z}_Y$ across the same input and observed a significant similarity improvement compared to token embeddings (As detailed in the Appendix A.1).

By aggregating discrete token embeddings into a single high-level embedding, we circumvent the discontinuities of token-level spaces. We thereby ensure that (1) manifold analysis is tractable (since $\mathbf{z}_X, \mathbf{z}_Y$ both lie in continuous $\mathbb{R}^d$ Mehta et al. (2019)), and (2) stochastic decoding does not cause major geometric shifts in these embeddings. This design choice, while straightforward, underpins all subsequent sections on manifold construction and robustness evaluation.

### 3.2 CONSTRUCTION OF MANIFOLDS VIA PGM

Understanding a language model's local robustness involves assessing how small input perturbations influence the model's output representation. A common strategy is to interpret this input–output mapping as a *manifold*, enabling geometric analyses of local stability (Rubin-Delanchy, 2020). However, directly constructing and maintaining such a manifold on raw embeddings can be both computationally and memory intensive. Recent work indicates that *graph-based* approaches can capture low-dimensional manifolds within high-dimensional data (Rubin-Delanchy, 2020), *especially* when the graph is constructed to preserve meaningful distances. PGMs (or Markov Random Fields) naturally encode these relationships in an undirected graphical structure, allowing for efficient inference about node neighborhoods and global structure (Koller, 2009). Specifically, Feng (2021) show that the graph structure learned by PGMs can approximate *resistance distances*, which in turn correlate with Euclidean distances among data samples. Hence, a properly built PGM manifold can reflect *both local and global* geometry—critical for analyzing small perturbations (local stability) and broader connectivity (global structure).

Despite their theoretical appeal, existing PGM-based methods rely on iterative optimization or dense computations (e.g., spectral factorization) that become prohibitive for large-scale graphs (Feng, 2021). When handling modern NLP datasets, where each sample might represent a document or prompt, and node counts can soar into the hundreds of thousands, these bottlenecks make traditional PGM approaches infeasible. To address scalability concerns, we propose a *near-linear complexity* method for building the PGM manifold. Intuitively, we seek a graph Laplacian structure (or precision matrix) that captures the intrinsic geometry of the reduced embeddings (Sec. 3.1) without incurring expensive global factorization steps. Previous work (Dong et al., 2019) shows that maximizing a penalized log-likelihood in the form of Equation. 1 yields a graph topology consistent with the underlying data distribution while preserving essential distance or similarity properties.

**PGM Objective.** Let $X \in \mathbb{R}^{|V| \times T}$ be the embedding matrix derived from Sec. 3.1, where each row corresponds to a sample. We aim to learn a precision matrix $\Theta$ that maximizes Dong et al. (2019):

$$\max_{\Theta} \quad F(\Theta) = \log \det(\Theta) - \frac{1}{k} \text{Tr}(X^{\top} \Theta X), \tag{1}$$

subject to $\Theta = \mathcal{L} + \frac{1}{\sigma^2} I$, where $\mathcal{L}$ is a valid Laplacian matrix and $\sigma^2$ is a prior variance term. Theorem 3.1 shows that *maximizing $F(\Theta)$ can be achieved using a spectral sparsification approach, which prunes edges with small distance ratios:

$$\rho_{p,q} = \frac{d^{\text{eff}}(p,q)}{d^{\text{dat}}(p,q)} = w_{p,q} \left( d^{\text{eff}}(p,q) \right),$$

where $d^{\text{eff}}(p,q)$ is the effective resistance distance (detailed in Appendix A.10) between nodes $p$ and $q$, $d^{\text{dat}}(p,q) = \|X_p - X_q\|_2^2$ is the data distance, and $w_{p,q} = 1/d^{\text{dat}}(p,q)$.

**Theorem 3.1.** *Maximizing the objective in Equation equation 1 can be done via an edge pruning strategy equivalent to spectral sparsification of the initial (dense) graph. Edges with small $\rho_{p,q}$ are removed, preserving the essential spectral (and thus resistance) properties of the original graph. The proof is available in Appendix A.2*

**Scalable Spectral Sparsification via Short-Cycle Decomposition.** A naive implementation of the above pruning requires frequent effective-resistance computations Spielman & Srivastava (2008), which is costly for weighted graphs. Methods such as short-cycle decomposition Chu et al. (2020) are effective for *unweighted* graphs but fail to retain accurate resistance distances when weights are discarded. We therefore introduce a refined *spectral sparsification* routine that uses low-resistance-diameter (LRD) decomposition to handle weighted edges without sacrificing the crucial resistance distance information.

**Theorem 3.2.** *Our LRD decomposition can compute the effective resistance of each edge and is capable of sparsifying weighted graphs. The proof is available in Appendix A.3*

**From PGM to Manifold.** With the pruned graph (and correspondingly updated Laplacian $\mathcal{L}$), solving Equation equation 1 yields a precision matrix $\Theta$ that encodes the desired topological relationships in $X$. This PGM thus underpins our low-dimensional manifold, accurately maintaining resistance distances for subsequent stability analyses (detailed in Appendix A.9). In practice, we initialize the graph with a $k$-nearest-neighbor construction and then apply our near-linear spectral sparsification (as detailed in Section 3.4) to achieve scalability. The resulting *manifold* reflects both local and global structures, enabling the DMD calculation.

## 3.3 Distance Mapping Distortion (DMD) Calculation

Having constructed the input and output manifolds (Section 3.2), we now introduce the DMD metric Cheng et al. (2021) to quantify a model's robustness at the *sample level*.

**Definition 3.3** (Distance Mapping Distortion (DMD)). Let $F$ be a function mapping an input manifold $G_X = (V, E_X)$ to an output manifold $G_Y = (V, E_Y)$, with $d_X(p, q)$ and $d_Y(p, q)$ denoting the distances between nodes $p$ and $q$ in $G_X$ and $G_Y$, respectively. The distance mapping distortion for $(p, q)$ through $F$ is

$$\gamma^F(p, q) \;=\; \frac{d_Y(p, q)}{d_X(p, q)}. \tag{2}$$

**Innovation Highlight:** We show that not only is $\gamma_{max}^F = \max_{p,q} \gamma^F(p, q)$ informative for worst-case local expansion, but also $(\gamma_{\min}^F)^{-1} = \left(\min_{p,q} \gamma^F(p, q)\right)^{-1}$ captures how the model might "collapse" distant inputs into overly similar outputs. We prove in Theorem 3.4 (below) that large $(\gamma_{\min}^F)^{-1}$ implies *another dimension* of poor robustness—distinct from $\gamma_{\max}^F$ (Empirical results are available in Appendix A.11). Hence, both extremes of the distortion spectrum are necessary for a full local analysis.

**Effective-resistance distance. (as detailed in Appendix A.10)** To make $\gamma^F$ computationally tractable, we replace geodesic distances with *effective-resistance* ($d^{eff}$). $d^{eff}(p, q)$ is always matched or upper-bounded by $d^{geo}(p, q)$ Chandra et al. (1996). Thus, $d^{eff}$ is an *efficient* surrogate for $d^{geo}$, especially when leveraging fast Laplacian solvers Koutis et al. (2010); Kyng & Sachdeva (2016). We then define

$$\gamma^F = \frac{d_Y^{eff}(p, q)}{d_X^{eff}(p, q)} = \frac{e_{p,q}^\top L_Y^+ e_{p,q}}{e_{p,q}^\top L_X^+ e_{p,q}}, \tag{3}$$

where $L_X$ and $L_Y$ denote the Laplacians of $G_X$ and $G_Y$, respectively. Computing $\gamma_{max}^F$ or $\gamma_{min}^F$ *exactly* via Equation equation 3 can still be expensive for large graphs, since it involves considering all node pairs $(p, q)$. To alleviate this, Cheng et al. (2021) proposed a *spectral* upper bound on $\gamma_{max}^F$, termed the $\lambda_{\max}(L_Y^+ L_X)$. Hence, a larger $\lambda_{\max}(L_Y^+ L_X)$ suggests a larger distortion ratio and thus poorer robustness. This is also the upper bound of the best Lipschitz constant under the manifold setting Cheng et al. (2021). For $\gamma_{min}^F$ lower bound calculation, we have:

**Theorem 3.4.** *The minimum distance mapping distortion $\gamma_{min}^F$ satisfies*

$$\gamma_{min}^F \;\geq\; \frac{1}{\lambda_{\max}(L_X^+ L_Y)}.$$

*The proof is available in Appendix A.4*

**SALMAN Score** measures, for each sample, how strongly the model distorts input–output distances, penalizing both expansion (near neighbors blown apart) and collapse (far points mapped too close), so larger scores indicate greater fragility. For each node (sample) $p$, we define **SALMAN**$^F(p)$:

$$\frac{1}{|\mathcal{N}(p)|} \sum_{q \in \mathcal{N}(p)} (\gamma^F(p,q)^3 + \gamma^F(p,q)^{-3}), \qquad (4)$$

where $\mathcal{N}_X(p)$ is the set of neighbors of $p$ in $G_X$ and $G_Y$. A node with a larger **SALMAN**$^F(p)$ is considered more *fragile*, since its local pairs $(p,q)$ exhibit greater distortion in "expansion" $(\gamma^F(p,q))$ or "collapse" $(1/\gamma^F(p,q))$ senses. To connect expansions and collapses more explicitly, let $\{\lambda_i\}_{i=1}^r$ be the $r$ largest eigenvalues of $L_Y^+ L_X$ with corresponding eigenvectors $\{v_i\}_{i=1}^r$, and let $\{\mu_i\}_{i=1}^r$ be the $r$ largest eigenvalues of $L_X^+ L_Y$ with corresponding eigenvectors $\{w_i\}_{i=1}^r$. We define the weighted eigensubspace matrices: $V_r = [v_1\sqrt{\lambda_1}, \ldots, v_r\sqrt{\lambda_r}]$, $W_r = [w_1\sqrt{\mu_1}, \ldots, w_r\sqrt{\mu_r}]$. For each pair $(p,q)$, one has:

**Theorem 3.5.** $\|W_r^\top e_{p,q}\|_2^2 + \|V_r^\top e_{p,q}\|_2^2 \propto \gamma^F(p,q)^3 + \gamma^F(p,q)^{-3}$. *The proof is available in Appendix A.5*

**Sample Selection and Correction.** Because SALMAN score is computed at the node or node-pair level, we can readily identify "high-distortion" samples and correct them via data augmentation or specialized re-training. This local approach complements global metrics, yielding a holistic robustness analysis pipeline.

### 3.4 COMPLEXITY

Our framework has *near-linear* time complexity with respect to the graph size. Below, we briefly outline the main steps and their costs. We first construct a $k$-NN graph from the data points (or embeddings) in $\mathbb{R}^d$. Using modern approximate nearest-neighbor algorithms (Malkov & Yashunin, 2018) with $O(|V| \log |V|)$. $|V|$ denotes the number of nodes in the graph. Then, We apply a Low-Resistance-Diameter (LRD) approach to sparsify the graph (Koutis et al., 2010; Cucuringu et al., 2016). Let $d$ be the average degree ( small in real-world graphs (Miao et al., 2019)) and $m$ be the dimension of a Krylov subspace. Then this step runs in $O(|V|\, d\, m)$, often simplified to $O(|V|\, m)$ under the sparse regime. Evaluating the SALMAN scores for all edges or nodes can be done in $O(|E|)$ time. $|E|$ denotes the number of edges in the graph. For sparse graphs with $|E| \approx d\,|V|$, this remains near-linear in $|V|$. Experimental results can be found in Appendix A.12.

## 4 EXPERIMENT RESULTS

We organize our experimental evaluation into three stages, each demonstrating how the *robustness ranking* (derived from the proposed SALMAN measure) can guide practical NLP tasks. The language models used for experiments range from BERT (136M), GPT-2 (1.5B) and the latest Llama3-8B. Details regarding data, hyperparameters, and model architectures are deferred to Appendix A.6

As this is the first work to propose a per-sample NLP robustness ranking, we lack direct comparisons with methods pursuing identical objectives. However, to address the lack of baseline concern, we compare SALMAN against: (1) Euclidean-distance-based ranking, which measures each sample's magnitude of embeddings without local manifold distortion; and (2) Jacobian-based sensitivity analysis. These baselines are simpler proxies for identifying "fragile" points. In Table 3, we show that both struggle to distinguish robust vs. non-robust samples under the same spaCy perturbation.

Additionally, we analyze representative robust versus non-robust samples to confirm that SALMAN reliably identifies vulnerable cases. Moreover, we compare our SALMAN measure against simpler baselines—such as random ranking, a state-of-the-art attack run without our approach, and a state-of-the-art robust training procedure without SALMAN—to assess whether our method provides tangible gains. Our experiments show that SALMAN surpasses these heuristic baselines on diverse perturbation benchmarks, offering strong evidence that SALMAN captures unique facets of sample-level vulnerability.

Table 1: Cosine similarities of robust and non-robust samples for different models on SST-2 and MNLI.

| Dataset | Model | Robust/Non-robust Cosine Similarity |
|---|---|---|
| SST-2 | BERT-base-uncased | 0.9911/0.8711 |
| | RoBERTa-base | 0.9992/0.9895 |
| | DistilBERT-base-uncased | 0.9955/0.9404 |
| | ALBERT-base-v2 | 0.9959/0.8279 |
| | GPT-2 | 0.9990 / 0.9153 |
| | LLaMA-7B-v2 | 0.9867 / 0.9160 |
| MNLI | BERT-base-uncased | 0.9902/0.9410 |
| | RoBERTa-base | 0.9993/0.9926 |
| | DistilBERT-base-uncased | 0.9971/0.9650 |
| | ALBERT-base-v2 | 0.9953/0.8709 |
| | GPT-2 | 0.9993 / 0.9904 |
| | LLaMA-7B-v2 | 0.9925 / 0.9842 |

Table 2: Robust and non-robust cosine similarities under two different attack methods (spaCy & TextAttack).

| Attack | Model | Dataset | Robust/Non-robust Cosine Sim. |
|---|---|---|---|
| spaCy | GPT-2 | SST-2 | 0.9981/0.9772 |
| | GPT-2 | MNLI | 0.9995/0.9730 |
| | LLaMA-7B-v2 | SST-2 | 0.9990/0.9751 |
| | LLaMA-7B-v2 | MNLI | 0.9825/0.9612 |
| TextAttack | GPT-2 | SST-2 | 0.9928/0.9413 |
| | GPT-2 | MNLI | 0.9945/0.9831 |
| | LLaMA-7B-v2 | SST-2 | 0.9548/0.8941 |
| | LLaMA-7B-v2 | MNLI | 0.9663/0.9479 |

Table 3: Cosine similarity between original vs. spaCy-perturbed samples on GPT-2, for Euclidean-, Jacobian-, and SALMAN-based rankings. We aim for robust sets to have higher similarity and non-robust sets to have lower similarity. SALMAN yields the largest gap.

| Method | SST-2 / MNLI | | Gap |
|---|---|---|---|
| Euclidean | R: 0.9953 / 0.9918 NR: 0.9986 / 0.9898 | $\rightarrow$ | 0.0033 / 0.0020 |
| Jacobian | R: 0.9964 / 0.9942 NR: 0.9965 / 0.9793 | $\rightarrow$ | 0.0001 / 0.0149 |
| SALMAN | R: 0.9981 / 0.9995 NR: 0.9772 / 0.9730 | $\rightarrow$ | **0.0209 / 0.0265** |

Table 4: BERTScore and KLD evaluations of robust vs. non-robust subsets (GPT-2, SST-2). Higher BERTScore indicates higher textual similarity. KLD measures distribution shift (lower is more stable).

| | KLD | BERTScore | | |
|---|---|---|---|---|
| | | Precision | Recall | F1 |
| Non-Robust | 0.1923 | 0.9961 | 0.9970 | 0.9965 |
| Robust | 7.6175e-07 | 0.9992 | 0.9991 | 0.9992 |

## 4.1 SAMPLE ROBUSTNESS EVALUATION

To validate that our robustness ranking meaningfully distinguishes between stable and fragile samples, we subject both *robust* (1% samples with lowest SALMAN) and *non-robust* (1% samples with highest SALMAN) samples to various NLP perturbations. These perturbations simulate natural edits or noise while controlling for the extent of modification via Levenshtein distance Ding et al. (2021). We thereby ensure that robust and non-robust subsets are perturbed equally in terms of edit cost, allowing a fair comparison of downstream output changes.

Following standard practices in text perturbations Guo et al. (2021); Ni et al. (2024), we implement three simple but widely used edits: deletion, insertion, and swap. Following previous works Le et al. (2022); Gupta et al. (2023); Jia et al. (2023), we measure the resultant output embedding shift via cosine similarity between the original and perturbed sentence embeddings, as seen in Table 1.

Beyond the three basic edits, we employ two state-of-the-art perturbation frameworks (spaCy Honnibal et al. (2020) and TextAttack Morris et al. (2020b)) to generate more sophisticated attacks. These methods leverage advanced synonym replacement and gradient-informed edits to produce challenging textual perturbations. Due to the substantial computational overhead of these approaches, we restrict them to two widely recognized LLMs—GPT-2 and LLaMA-7B-v2—thereby striking a balance between experimental rigor and resource feasibility. In Table 2, we apply each SOTA method to both robust and non-robust subsets, measuring the resulting cosine similarities to assess susceptibility to adversarial manipulations.

To further assess the difference between robust and non-robust samples, we incorporate two additional metrics: KL-Divergence (KLD) and BERTScore Zhang et al. (2019b). Table 4 shows that non-robust samples exhibit larger distribution shift (higher KLD) and lower textual similarity (BERTScore) under perturbations, whereas robust samples remain highly similar. We provide statistical reliability analysis in Appendix A.1.

By systematically perturbing both robust and non-robust samples, we confirm that non-robust samples consistently exhibit greater output variability under identical input changes. This aligns with prior evidence that local text modifications can disproportionately affect certain data points Morris et al. (2020a), and it underscores the value of distance mapping distortion in identifying vulnerabilities at the sample level.

## 4.2 SALMAN-GUIDED ATTACK

Jailbreak Attacks are an important way to assess the security and robustness of LLM Yi et al. (2024); Chu et al. (2024). By strategically crafting prompts, it is possible to bypass the LLM's inherent safeguards and generate harmful content. Recently, numerous studies have focused on automatically generating stealthy jailbreak attack prompts. However, these current methods are both labor-intensive and computationally demanding. We propose using the SALMAN score to guide more effective attacks.

**Motivation: Find the Non-Robust Data Samples.** We harness the robustness ranking to focus adversarial efforts on the most susceptible samples. This strategy is akin to reducing query complexity in black-box attacks or prioritizing the most "fragile" points. We structured the experiment as follows: 1) we rank the dataset by descending SALMAN score (i.e., from least robust to most robust). 2) We perform the existing attack method *only* on the top $k\%$ of non-robust samples. 3) Then we randomly sample $k\%$ data and use the same method to attack LLM again as a comparison.

For our experiment, we take GCG Zou et al. (2023) and AutoDAN Liu et al. (2023) as the jailbreak attack method and use the AdvBench Harmful Behaviors dataset Zou et al. (2023) to evaluate the jailbreak attacks. This dataset has 520 data points (Dataset detail in Appendix A.6). After ranking all the data using SALMAN, we selected the top 1% of unstable samples to launch attacks on LLMs, supplemented by randomly sampling another 1% of the samples for the same purpose. Subsequently, we evaluated the effectiveness of SALMAN by comparing changes in the Attack Success Rate (ASR) and the number of attack attempts (Steps). We also justify extracting embeddings from the language model's first and final hidden layers, which capture complementary semantic information (Empirical results are available in Appendix A.13.2).

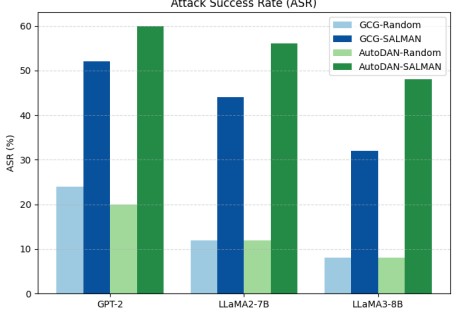 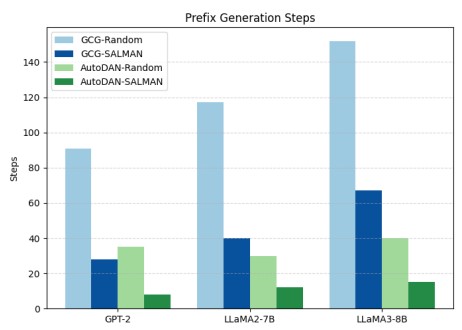

(a) Attack success rate (ASR) and prefix generation steps across different models and attack methods.

(b) Attacking efficiency: the comparison of the number of attacking attempts.

Figure 2: Comparison of adversarial attack performance (a) and efficiency (b) with and without SALMAN-guided selection.

Figure 2 (a) shows that attacking these low-robustness samples first yields higher success rates *and* reduced time-to-attack compared to random sampling baselines. We visualized attacking efficiency in Figure 2 (b). The SALMAN-based ranking serves as an efficient "shortcut" for adversarial testing. Then, we investigate SALMAN ranking by selecting different top-$k$ percentages of the dataset. We bin (10%) the entire dataset into deciles by SALMAN rank and apply the same attack methods to each bin. As $k$ increases, we include more (relatively) robust samples, resulting in lower overall ASR and efficiency. We further assessed the SALMAN-Guided attack using proxy models to evaluate the robustness of the proposed method. The experimental results are presented in Table 5. The results demonstrate that SALMAN-guided attacks retain high effectiveness across models, confirming their transferability. We also conduct the SALMAN-guided attack experiment on the multilingual Jailbreak dataset and we show the results in Appendix A.13.

## 4.3 SALMAN-GUIDED LLM FINE-TUNING

Fine-tuning LLMs sometimes leads to overfitting, losing key representations from pre-training Howard & Ruder (2018); Neerudu et al. (2023). Prior work has attempted to measure how much an LLM's internal representations drift from the pre-trained checkpoint, using similarity metrics such as CKA or STIR Neerudu et al. (2023). A large drift often indicates the model is overfitting, thus sacrificing generalization and robustness in practice.

Table 5: Proxy-based SALMAN ranking vs. target LLM under GCG. We list the ASR on the top-1% non-robust subsets identified by the proxy model.

| Proxy | GPT-2 | LLaMA2-7B | LLaMA3-8B |
|---|---|---|---|
| GPT-2 | 60% | 48% | 40% |
| LLaMA2-7B | 60% | 56% | 40% |
| LLaMA3-8B | 60% | 56% | 48% |

Table 6: Performance of ROSE fine-tuning on SST-2, RTE, and QNLI tasks with RoBERTa$_{BASE}$. Each cell shows *ROSE fine-tuning / SALMAN-guided ROSE fine-tuning* accuracy. Bold is better.

| Task | GLUE | AdvGLUE |
|---|---|---|
| SST-2 | 94.84 / **94.84** | 37.67 / **41.22** |
| RTE | 78.34 / **78.34** | 35.49 / **40.75** |
| QNLI | 92.19 / **92.81** | 44.19 / **48.02** |

Table 7: Model-level robustness score Neerudu et al. (2023) of BERT and GPT-2 on CoLA and SST-2 under various perturbations. Each cell shows the *Normal/SALMAN-guided* fine-tuning value. Better results are in **bold**.

| Perturbation | CoLA | SST-2 |
|---|---|---|
| | *BERT* | |
| Drop nouns | 0.18 / **0.29** | 0.92 / **1.02** |
| Drop verbs | 0.05 / **0.22** | 0.95 / **1.03** |
| Drop first | 0.48 / **0.70** | 0.98 / **1.01** |
| Drop last | 0.34 / **0.72** | 1.00 / **1.00** |
| Swap text | 0.13 / **0.22** | 0.98 / **1.01** |
| Add text | 0.85 / **0.88** | 0.99 / **0.99** |
| Change char | 0.14 / **0.24** | 0.84 / **0.91** |
| Bias | 0.95 / **0.99** | 1.00 / **1.00** |
| | *GPT-2* | |
| Drop nouns | 0.10 / **0.47** | 0.93 / **1.00** |
| Drop verbs | 0.24 / **0.32** | 0.95 / **1.00** |
| Drop first | 0.75 / **0.91** | 0.97 / **1.01** |
| Drop last | 0.45 / **0.78** | 0.99 / **1.01** |
| Swap text | 0.16 / **0.45** | 0.98 / **1.00** |
| Add text | 0.92 / **0.96** | 0.99 / **1.01** |
| Change char | 0.29 / **0.36** | 0.86 / **1.02** |
| Bias | 0.96 / **1.14** | **1.01** / 0.99 |

**Motivation: Focus on Non-Robust Data.** Several studies show that focusing on non-robust samples during training can improve model robustness and generalizability (Cheng et al., 2021; Zhu et al., 2023a). Inspired by these findings, we propose to *down-weight* robust samples and *up-weight* non-robust samples (as determined by our SALMAN-based ranking) when fine-tuning an LLM. The intuition is that easy/robust data rarely contributes to boosting generalizable features, whereas harder (high DMD) data pushes the model to learn more discriminative patterns.

We follow the fine-tuning protocol described by Neerudu et al. (2023). By placing greater emphasis on non-robust data (as detailed in Appendix A.8) , we hypothesize that the fine-tuned model retains more generalizable features from its pre-training, avoiding over-specialization to easy examples. We observe two key outcomes: *1) Comparable Performance, Closer to Pre-training:* SALMAN-guided LLMs achieve comparable or better accuracy vs. standard fine-tuning, yet exhibit higher similarity to the pre-trained checkpoint. On GLUE tasks, we find up to 54% gains in CKA or STIR, signifying less drift with better accuracy. Results appear in Appendix A.7 and highlight key findings. *2) Enhanced Robustness Scores:* Although our paper introduces a *sample-centric* robustness measure for *ranking* individual samples *without* requiring explicit perturbations, we also need to assess the *entire model*'s robustness after fine-tuning. To this end, we adopt the model-level robustness score proposed by Neerudu et al. (2023), which measures how representations change under various text perturbations across the full dataset. Measuring each model's robustness scores confirms that the *SALMAN-guided* LLM obtains higher robustness than a conventionally fine-tuned model. As shown in Table 7, we attribute this improvement to the heightened focus on challenging (non-robust) samples during training.

**Combining with SOTA Robust Training.** We further integrate our approach with ROSE (Jiang et al., 2022), a selective fine-tuning framework that prunes "spurious" parameter updates to achieve greater adversarial resilience. Specifically, we embed our SALMAN-based weighting (as detailed in Appendix A.8) within ROSE's parameter selection process. Experimental results in Table 6 reveal that sample-level weighting and parameter-level selection are complementary strategies.

## 5 CONCLUSION

We introduced SALMAN, a novel measure to identify and rank the local robustness of transformer-based language models. Our experiments across diverse benchmarks and large language models reveal that SALMAN not only distinguishes *robust* from *non-robust* samples under both simple and SOTA perturbations, but also effectively guides attacks and fine-tuning. Moreover, incorporating SALMAN into the existing robust training framework yields even greater resilience against adversarial perturbations. These results underscore the potential of leveraging *sample-level robustness* to bolster both attack strategies and robust model adaptation.

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

## A APPENDIX

### A.1 DETERMINISTIC HIDDEN-STATE EMBEDDINGS

Though modern transformers can produce *stochastic* outputs at the token level (e.g., due to beam search, random sampling, or dropout), their internal hidden states can remain *largely deterministic* under fixed conditions (Wolf, 2020). Below, we validate this claim by comparing token-level embeddings and pooled hidden-state embeddings across different decoding strategies. We then observe what happens when we additionally fix the seed.

**Token vs. Pooled Embeddings Under Varying Seeds.** We feed the *same* input sequence through various transformers (DistilBERT, BERT, RoBERTa, and Google-Electra), each time *without* enforcing a fixed random seed for decoding. We then collect:

- Token Embeddings. The final output embeddings for each token in the decoded sequence (i.e., after language modeling head).
- Pooled MHSA Output Embeddings. Our approach aggregates multiple attention heads and pools them into a single output vector per sequence, thereby abstracting away token-level variations.

For each model, we compute the cosine similarity between embeddings arising from different decoding runs of the *same* input. Table 8 shows representative results for three datasets: SQuAD, IMDB, and AG-News.

Table 8: Cosine Similarity of Embeddings Across Different Decoding Runs *Without* a Fixed Random Seed. Higher is more stable.

| | Token Embeddings | | | |
| --- | --- | --- | --- | --- |
| Dataset | DistilBERT | BERT | RoBERTa | Google-Electra |
| SQuAD | 0.9338 | 0.2302 | 0.9967 | 0.1859 |
| IMDB | 0.9685 | 0.4443 | 0.9977 | 0.5131 |
| AG-News | 0.9450 | 0.5967 | 0.9954 | 0.5771 |

| | Pooled MHSA Output Embeddings | | | |
| --- | --- | --- | --- | --- |
| Dataset | DistilBERT | BERT | RoBERTa | Google-Electra |
| SQuAD | 1.00 | 1.00 | 1.00 | 1.00 |
| IMDB | 1.00 | 1.00 | 1.00 | 1.00 |
| AG-News | 0.99 | 0.99 | 0.99 | 0.99 |

Even when seeds vary, **token-level embeddings** exhibit inconsistent cosine similarity across runs (e.g., BERT scoring only 0.23 for SQuAD). By contrast, **our pooled MHSA method** maintains consistently high similarity (0.99 or 1.00), indicating a more stable representation that does not fluctuate with token-level decoding choices. This stability suggests that the representation potentially captures a more consistent global semantic space Reimers (2019).

**Fixed Seed + Pooled MHSA**    Finally, we fix the random seed (and disable dropout) for all runs using our pooled MHSA approach, ensuring the only factor causing embedding changes is an explicit *input* perturbation (e.g., synonyms swapped). Under *identical* inputs and the same seed, the pooled MHSA output embeddings always match exactly (cosine similarity = 1.00), regardless of how tokens might be sampled. As summarized in Table 9, **all entries become** 1.00 when there is *no input perturbation*.

Table 9: Cosine Similarity with Fixed Seed *and* Pooled MHSA. Identical inputs yield identical embeddings (similarity = 1.00).

|  | Seed-Fixed Pooled MHSA Output | | | |
|---|---|---|---|---|
| Dataset | DistilBERT | BERT | RoBERTa | Google-Electra |
| SQuAD | 1.00 | 1.00 | 1.00 | 1.00 |
| IMDB | 1.00 | 1.00 | 1.00 | 1.00 |
| AG-News | 1.00 | 1.00 | 1.00 | 1.00 |

**Statistical Reliability Analysis**    We consistently fixed random seeds to minimize variability and ensure reproducibility. To further quantify the statistical stability of SALMAN rankings, we conducted multi-run experiments on the SST-2 dataset (over 69k samples, the largest dataset in our study). The results reveal a very high degree of consistency, with a Top-20% overlapping rate of $99.0\% \pm 1.2\%$, underscoring the statistical reliability of the SALMAN ranking. Moreover, we repeated the top-1% non-robustness ranking five times and evaluated the subsets using AutoDAN (under a time-limited setting). The attack success rates were 80%, 60%, 60%, 60%, and 60%, respectively. These results provide additional empirical evidence that SALMAN rankings are not only statistically stable but also practically reliable for guiding adversarial attacks.

A.2    PROOF FOR THEOREM 3.1

We now show that *maximizing* the objective

$$\max_{\Theta} \quad F(\Theta) = \log\det(\Theta) - \frac{1}{k}\mathrm{Tr}(X^{\top}\Theta X), \tag{5}$$

where $\Theta = \mathcal{L} + \frac{1}{\sigma^2}I$, can be achieved by removing (or down-weighting) edges whose *distance ratio* is small. In essence, these low-ratio edges contribute less to $\log\det(\Theta)$ while incurring a larger penalty in the trace term, so pruning them increases $F(\Theta)$.

**1.  Decomposing the Objective.**    Writing $\mathcal{L} = \sum_{(p,q)\in E} w_{p,q}\, e_{p,q}\, e_{p,q}^{\top}$, we split $F(\Theta)$ into two terms:

$$F(\Theta) = F_1(\Theta) - \frac{1}{k}F_2(\Theta), \quad \text{where}$$

$$F_1(\Theta) = \log\det(\Theta), \quad F_2(\Theta) = \mathrm{Tr}(X^{\top}\Theta X).$$

Since $\Theta = \mathcal{L} + \frac{1}{\sigma^2}I$, each edge weight $w_{p,q}$ appears explicitly in $\mathcal{L}$.

**2.  Gradient with Respect to an Edge Weight.**    To optimize $F(\Theta)$ w.r.t. a single edge weight $w_{p,q}$:

- **Term** $F_1(\Theta)$: Let $\lambda_i$ be the $i$-th eigenvalue of $\mathcal{L}$, and $v_i$ its eigenvector. Then

$$\frac{\partial}{\partial w_{p,q}}\Big(\log\det(\Theta)\Big) = \frac{\partial}{\partial w_{p,q}}\Big[\log\det\big(\mathcal{L} + \frac{1}{\sigma^2}I\big)\Big].$$

By standard matrix calculus, this derivative can be linked to the *effective resistance distance* $d^{\mathrm{eff}}(p,q)$:

$$\frac{\partial F_1}{\partial w_{p,q}} \;\approx\; d^{\mathrm{eff}}(p,q),$$

where $d^{\mathrm{eff}}(p,q)$ encapsulates how strongly edge $(p,q)$ influences $\mathrm{logdet}(\Theta)$.

- **Term $F_2(\Theta)$:**

$$F_2(\Theta) = \mathrm{Tr}\big(X^\top \Theta X\big) = \mathrm{Tr}\big(X^\top (\mathcal{L} + \tfrac{1}{\sigma^2} I)\, X\big) = \frac{\mathrm{Tr}(X^\top X)}{\sigma^2} + \sum_{(p,q)\in E} w_{p,q} \,\big\|X^\top e_{p,q}\big\|_2^2.$$

Since $\big\|X^\top e_{p,q}\big\|_2^2 = \|X_p - X_q\|_2^2 = d^{\mathrm{dat}}(p,q)$, we have

$$\frac{\partial F_2}{\partial w_{p,q}} \;=\; \big\|X_p - X_q\big\|_2^2 \;=\; d^{\mathrm{dat}}(p,q).$$

Furthermore, $d^{\mathrm{dat}}(p,q) = \frac{1}{w_{p,q}}$, which implies

$$\frac{\partial F_2}{\partial w_{p,q}} \;=\; \frac{1}{w_{p,q}}.$$

Hence, the derivative of $F(\Theta) = F_1 - \frac{1}{k} F_2$ w.r.t. $w_{p,q}$ is

$$\frac{\partial F}{\partial w_{p,q}} \;=\; d^{\mathrm{eff}}(p,q) \;-\; \frac{1}{k}\,\frac{1}{w_{p,q}}. \tag{6}$$

**3. Distance Ratio and Pruning Condition.**   Rewriting Equation equation 6:

$$d^{\mathrm{eff}}(p,q) \;-\; \frac{1}{k}\,\frac{1}{w_{p,q}} = 0 \iff d^{\mathrm{eff}}(p,q) \;=\; \frac{1}{k}\,\frac{1}{w_{p,q}}.$$

Define the *distance ratio* for edge $(p,q)$:

$$\rho_{p,q} = \frac{d^{\mathrm{eff}}(p,q)}{d^{\mathrm{dat}}(p,q)} \;=\; w_{p,q}\,\big(d^{\mathrm{eff}}(p,q)\big).$$

When $d^{\mathrm{eff}}(p,q)$ is relatively large compared to $\frac{1}{w_{p,q}}$, we have $\rho_{p,q}$ large, indicating an important edge for $\mathrm{logdet}(\Theta)$. Conversely, if $\rho_{p,q}$ is small, the edge $(p,q)$ contributes little to $F_1(\Theta)$ but increases $F_2(\Theta)$, thereby reducing $F(\Theta)$.

**4. Conclusion: Prune Low-Ratio Edges.**   Thus, maximizing equation 5 naturally leads to *removing or down-weighting* edges whose ratio

$$\rho_{p,q} \;=\; \frac{d^{\mathrm{eff}}(p,q)}{d^{\mathrm{dat}}(p,q)}$$

is below a certain threshold. By pruning these edges, we preserve the essential spectral structure needed to keep $\mathrm{logdet}(\Theta)$ high (reflecting higher effective resistance) while mitigating the penalty in $\mathrm{Tr}(X^\top \Theta X)$ from edges that have large data distance but small effective resistance. In other words, *edges with large $\rho_{p,q}$ stay*, and edges with small $\rho_{p,q}$ are pruned, thereby maximizing $F(\Theta)$ and maintaining the key (Laplacian) properties of the original graph.

### A.3 PROOF OF LRD DECOMPOSITION FOR EFFICIENT EDGE RESISTANCE COMPUTATION AND WEIGHTED GRAPH SPARSIFICATION

In this section, we establish that the *low-resistance-diameter* (LRD) decomposition scheme can efficiently approximate the effective resistance for each edge in a *weighted* graph and thus provide an effective path toward spectral sparsification. Our argument proceeds in two stages:

1. **Approximate Effective Resistance via Krylov Subspaces:** We show how the iterative procedure yields reliable estimates of $d^{\mathrm{eff}}(p,q)$ in near-linear time.

2. **Bound Cycle Lengths under LRD Clustering:** We explain how the multilevel contraction and supernode formation ensure that edges with large resistance distances are effectively sampled or retained, yielding a final sparsified graph that spectrally approximates the original.

**Stage 1: Approximating Effective Resistances via Krylov Subspaces.** The resistance distance for an edge $(p, q)$ in a graph $G = (V, E)$ with Laplacian $L_G$ can be expressed as:

$$d^{\text{eff}}(p, q) = \sum_{i=2}^{N} \frac{\left(u_i^{\top} e_{p,q}\right)^2}{u_i^{\top} L_G\, u_i},$$

where $u_2, \ldots, u_N$ are the (nontrivial) eigenvectors of $L_G$ and $e_{p,q} = e_p - e_q$. Directly computing all eigenvalues/eigenvectors for large $G$ is typically prohibitive. Instead, we replace these eigenvectors with a small set of Krylov basis vectors $x^{(1)}, x^{(2)}, \ldots, x^{(m)}$, which approximate the subspace spanned by the top spectral components of $L_G$. Specifically, each $x^{(i)}$ is drawn from

$$\kappa_m(A, c) = \operatorname{span}\{\, c,\, A\,c,\, A^2\,c,\, \ldots,\, A^{m-1}\,c\,\},$$

where $A$ is the adjacency matrix and $c$ is a random vector. Orthogonalizing and normalizing these $m$ vectors ensures a concise basis in which to project $e_{p,q}$.

**Lemma A.1** (Krylov Approximation of Effective Resistance). *Suppose $x^{(1)}, \ldots, x^{(m)}$ are $m$ orthonormal vectors approximating the dominant spectral subspace of $L_G$ (via a Krylov process). Then for any edge $(p, q) \in E$,*

$$d^{\text{eff}}(p, q) \approx \sum_{i=1}^{m} \frac{\left(x^{(i)\top} e_{p,q}\right)^2}{x^{(i)\top} L_G\, x^{(i)}}.$$

*Choosing $m = \widetilde{O}(\log N)$ and updating each level in near-linear time yields high-probability error bounds comparable to exact spectral decompositions (Spielman & Srivastava, 2011; Koutis et al., 2010).*

**Stage 2: LRD-based Short-Cycle Decomposition for Weighted Graphs.** The second key step is the *multilevel* contraction scheme that ensures edges with large effective resistance remain "visible" at higher levels, while short cycles (or low-resistance edges) are contracted to form supernodes. Specifically:

- At level $\delta$, each edge $(p, q)$ is either *contracted* (if $d_{eff}^{(\delta)}(p, q)$ is below the chosen threshold) or *retained* (if $d_{eff}^{(\delta)}(p, q)$ is above the threshold). Contraction merges $p$ and $q$ into a supernode $\vartheta$, assigning it an accumulated weight $\eta_\vartheta$ via:

$$\eta_\vartheta := \eta(p^{(\delta)}) + \eta(q^{(\delta)}) + d_{eff}^{(\delta)}(p, q). \tag{7}$$

- As edges are contracted, any cycles formed at level $\delta$ have length bounded by the effective-resistance diameter. Consequently, short cycles in the *weighted* setting are handled similarly to Chu et al. (2020)'s unweighted approach, except that we measure distances via $d_{eff}$, not just hop counts.

- After finalizing the clusters (supernodes), the "inter-cluster" edges (those bridging different clusters) are preserved or upweighted in the sparsified graph. These edges typically have higher $d_{eff}(p, q)$ and thus significantly impact spectral properties of $L_G$.

Formally, let $L_H$ denote the Laplacian of the sparsified graph $H$ returned by the LRD decomposition. We say $H$ is a $(1 \pm \varepsilon)$-*spectral-approximation* of $G$ if, for all $x \in \mathbb{R}^N$,

$$(1 - \varepsilon)\, x^{\top} L_G\, x \;\leq\; x^{\top} L_H\, x \;\leq\; (1 + \varepsilon)\, x^{\top} L_G\, x.$$

Standard arguments from spectral sparsification (Spielman & Srivastava, 2011) show that any procedure ensuring accurate effective-resistance estimates can preserve the graph's quadratic form up to $(1 \pm \varepsilon)$ factors. The main difference in *our* approach is the use of low-resistance-diameter cycles instead of purely unweighted short cycles.

**Theorem A.2** (LRD for Weighted Graph Sparsification). *Let $G = (V, E)$ be a connected weighted graph with $N$ nodes and $M$ edges, and let $0 < \varepsilon < 1$ be a chosen approximation factor. Then, by applying the LRD-based spectral sparsification algorithm with Krylov-based effective-resistance estimates:*

1. *We can approximate $d^{\text{eff}}(p, q)$ for all edges $(p, q) \in E$ in near-linear time (Lemma A.1).*

2. *We contract short cycles (below a chosen $d_{eff}$ threshold) and preserve inter-cluster edges with high $d_{eff}(p, q)$, forming a sparsified graph $H$ with Laplacian $L_H$.*

3. *With high probability, $H$ satisfies $(1 - \varepsilon)\, x^\top L_G x \;\leq\; x^\top L_H x \;\leq\; (1 + \varepsilon)\, x^\top L_G x, \; \forall x \in \mathbb{R}^N$.*

*Hence, $H$ serves as a $(1 \pm \varepsilon)$-spectral-approximation to $G$, yielding a low-complexity graph that closely preserves the original graph's spectral (and thus effective-resistance) structure.*

*Proof Sketch.*

*(1) Effective-resistance approximation.* By Lemma A.1, each edge's resistance can be estimated via $m = \widetilde{O}(\log N)$ Krylov vectors per level of the hierarchy. Summed over $\delta$ levels, the total cost remains near-linear in $N + M$ (plus polylogarithmic factors), similar to Koutis et al. (2010); Kyng & Sachdeva (2016).

*(2) Cycle decomposition.* Following Chu et al. (2020), short cycles are identified and contracted; we adapt the criteria to *resistance distances* in lieu of unweighted hop distances. The LRD threshold ensures each supernode aggregates edges that have sufficiently low $d_{eff}(p, q)$, while edges with higher $d_{eff}(p, q)$ remain across clusters and are re-inserted (or re-weighted) in the final sparsified graph $H$.

*(3) Spectral approximation.* Standard spectral graph theory arguments Spielman & Srivastava (2011) show that removing or down-weighting edges of low effective resistance induces little change in $x^\top L_G x$ for all $x$. Conversely, preserving edges with large $d_{eff}(p, q)$ is crucial for maintaining the spectral signature of $L_G$. The result is a $(1 \pm \varepsilon)$-approximation for sufficiently small $\varepsilon$.

Thus, LRD-based cycle decomposition extends short-cycle approaches to weighted graphs by anchoring cycle lengths in *resistance* metrics. This achieves the final $(1 \pm \varepsilon)$ spectral-approximation for $G$ in near-linear time. $\qquad\square$

## A.4 Proof for the Relationship Between $\gamma_{min}^F$ and $\lambda_{\max}(L_X^+ L_Y)$

**Definition A.3** (Minimum Distance Mapping Distortion). Analogous to the definition of $\gamma_{max}^F$, we define

$$\gamma_{min}^F \;=\; \min_{\substack{p,\, q \in V \\ p \neq q}} \frac{d_Y(p, q)}{d_X(p, q)} \;=\; \min_{\substack{p,\, q \in V \\ p \neq q}} \frac{e_{p,q}^\top L_Y^+ e_{p,q}}{e_{p,q}^\top L_X^+ e_{p,q}},$$

where $L_X^+$ and $L_Y^+$ are the Moore–Penrose pseudoinverses of the Laplacian matrices $L_X$ and $L_Y$, respectively. This quantity reflects the *smallest* ratio of output distance to input distance, characterizing how close points in the *output* manifold might originate from distant points in the *input* manifold.

*Proof.* Recall that

$$\gamma_{min}^F \;=\; \min_{\substack{p,\, q \in V \\ p \neq q}} \frac{e_{p,q}^\top L_Y^+ e_{p,q}}{e_{p,q}^\top L_X^+ e_{p,q}}.$$

If we define $v = e_{p,q}$ and restrict $v$ such that $v^\top \mathbf{1} = 0$ ($\mathbf{1}$ being the all-ones vector, ensuring we stay within the subspace on which the Laplacian pseudoinverses are invertible), then

$$\gamma_{min}^F \;\geq\; \min_{\substack{\|v\| \neq 0 \\ v^\top \mathbf{1} = 0}} \frac{v^\top L_Y^+ v}{v^\top L_X^+ v}.$$

By the (min-)max version of the generalized Courant-Fischer theorem (applied to positive semidefinite matrices on the subspace orthogonal to $\mathbf{1}$), we have

$$\min_{\substack{\|v\| \neq 0 \\ v^\top \mathbf{1} = 0}} \frac{v^\top L_Y^+ v}{v^\top L_X^+ v} \;=\; \lambda_{\min}\!\left(L_Y^+ L_X\right).$$

However, because $L_Y^+ L_X$ is invertible on that same subspace, one also obtains the relationship

$$\lambda_{\min}\left(L_Y^+ L_X\right) \;=\; \frac{1}{\lambda_{\max}\left(\left(L_Y^+ L_X\right)^{-1}\right)}.$$

Next, it can be shown that

$$\left(L_Y^+ L_X\right)^{-1} \;=\; L_X^+ L_Y \quad \text{(on the subspace } v^\top \mathbf{1} = 0),$$

which gives

$$\lambda_{\min}\left(L_Y^+ L_X\right) \;=\; \frac{1}{\lambda_{\max}\left(L_X^+ L_Y\right)}.$$

Combining these steps, we conclude

$$\gamma_{min}^F \;\geq\; \lambda_{\min}\left(L_Y^+ L_X\right) \;=\; \frac{1}{\lambda_{\max}\left(L_X^+ L_Y\right)}.$$

This completes the proof. $\qquad\square$

### A.5 SALMAN SCORE AND CORRESPONDING PROOFS

Specifically, we first compute the weighted eigensubspace matrix $V_r \in \mathbb{R}^{N \times r}$ for spectral embedding on $G_X$ with $N$ nodes:

$$V_r \overset{\text{def}}{=} \left[ v_1 \sqrt{\lambda_1}, ..., v_r \sqrt{\lambda_r} \right], \tag{8}$$

where $\lambda_1, \lambda_2, ..., \lambda_r$ represent the first $r$ largest eigenvalues of $L_Y^+ L_X$ and $v_1, v_2, ..., v_r$ are the corresponding eigenvectors. Let $u_1, u_2, ..., u_N$ denote the $N$ eigenvectors of $L_X L_Y^+$, respectively, while their corresponding eigenvalues are shared. In addition, eigenvectors $u_i$ can be constructed to satisfy:

$$u_i^\top L_X^+ u_j = \begin{cases} 1, & i = j \\ 0, & i \neq j. \end{cases} \tag{9}$$

$$\Rightarrow u_i^\top L_Y^+ u_j = \begin{cases} \lambda_i, & i = j \\ 0, & i \neq j. \end{cases} \tag{10}$$

Therefore, the following equations hold:

$$L_Y^+ u_i = \lambda_i L_X^+ u_i \leftrightarrow L_Y^+ L_X \left(L_Y^+ u_i\right) = \lambda_i \left(L_Y^+ u_i\right)$$
$$L_X v_i = \lambda_i L_Y v_i \leftrightarrow L_Y^+ L_X v_i = \lambda_i v_i \tag{11}$$

which leads to the following equation

$$v_i = \beta_i L_Y^+ u_i$$

$$\Rightarrow u_j^\top v_i = \begin{cases} \beta_i \lambda_i, & i = j \\ 0, & i \neq j. \end{cases} \tag{12}$$

where $\beta_i$ denotes a scaling coefficient. Without loss of generality, $e_{p,q}$ can be expressed as a linear combination of $u_i$ for $i = 1, ..., N$ as follows:

$$e_{p,q} = \sum_{i=1}^{N} \alpha_i u_i. \tag{13}$$

Then $\gamma^F(p, q)$ can be rewritten as follows:

$$\begin{aligned}
\gamma^F(p, q) &= \frac{d_Y(p, q)}{d_X(p, q)} = \frac{e_{p,q}^\top L_Y^+ e_{p,q}}{e_{p,q}^\top L_X^+ e_{p,q}} \\
&= \frac{\left(\sum_{i=1}^{N} \alpha_i u_i\right)^\top L_Y^+ \left(\sum_{i=1}^{N} \alpha_i u_i\right)}{\left(\sum_{i=1}^{N} \alpha_i u_i\right)^\top L_X^+ \left(\sum_{i=1}^{N} \alpha_i u_i\right)} \\
&= \frac{\sum_{i=1}^{N} \sum_{j=1}^{N} \alpha_i \alpha_j u_i^\top L_Y^+ u_j}{\sum_{i=1}^{N} \sum_{j=1}^{N} \alpha_i \alpha_j u_i^\top L_X^+ u_j} \\
&= \frac{\sum_{i=1}^{N} \alpha_i^2 u_i^\top L_Y^+ u_i}{\sum_{i=1}^{N} \alpha_i^2 u_i^\top L_X^+ u_i} \\
&= \frac{\sum_{i=1}^{N} \alpha_i^2 \lambda_i}{\sum_{i=1}^{N} \alpha_i^2}.
\end{aligned} \tag{14}$$

If the edge $(p, q)$ is dominantly aligned with a single dominant generalized eigenvector $u_k$ where $1 \leq k \leq r$, it implies $\forall i \neq k, \alpha_i \approx 0$ and thus $e_{p,q} \approx \alpha_k u_k$. Then:

$$\gamma^F(p, q) \approx \lambda_k. \tag{15}$$

With $\left\| V_r^\top e_{p,q} \right\|_2^2$, We have:

$$
\begin{aligned}
\|V_r^\top e_{p,q}\|_2^2 &= \sum_{i=1}^{r} \lambda_i (v_i^\top e_{p,q})^2 \\
&= \sum_{i=1}^{r} \lambda_i \left( \sum_{j=1}^{N} \alpha_j \beta_i u_j^\top L_Y^+ u_i \right)^2 \\
&= \sum_{i=1}^{r} \alpha_i^2 \beta_i^2 \lambda_i^3 \\
&\approx \alpha_k^2 \beta_k^2 \lambda_k^3 \propto \left( \gamma^F(p, q) \right)^3
\end{aligned}
\tag{16}
$$

Consider $L_X^+ L_Y$ whose eigenvalues we denote by $\mu_1, \ldots, \mu_N$ with corresponding eigenvectors $w_1, \ldots, w_N$. we then compute the weighted eigensubspace matrix $W_r \in \mathbb{R}^{N \times r}$ for spectral embedding on $G_Y$ with $N$ nodes:

$$W_r \stackrel{\text{def}}{=} [w_1 \sqrt{\mu_1}, ..., w_r \sqrt{\mu_r}], \tag{17}$$

Because $L_X^+ L_Y$ has eigenvalues $\mu_i = 1/\lambda_i$, and its eigenvectors $w_i$ correspond in a reciprocal way, one obtains a parallel statement. In particular:

- If $e_{p,q}$ aligns chiefly with the eigenvector $w_k$ of $L_X^+ L_Y$ having eigenvalue $\mu_k = 1/\lambda_k$,
- Then $\gamma^F(p, q) \approx \lambda_k$ as before.

A similar calculation to the Equation 16 proof now yields

$$\left\| W_r^\top e_{p,q} \right\|_2^2 = \sum_{i=1}^{r} \mu_i \left( w_i^\top e_{p,q} \right)^2 \approx \text{const} \times (\mu_k)^3 = \text{const} \times \left( \frac{1}{\lambda_k} \right)^3.$$

Since $\lambda_k \approx \gamma^F(p, q)$, we conclude

$$\left\| W_r^\top e_{p,q} \right\|_2^2 \propto \gamma^F(p, q)^{-3}.$$

Hence, $\left\| W_r^\top e_{p,q} \right\|_2^2 + \left\| V_r^\top e_{p,q} \right\|_2^2 \propto \gamma^F(p, q)^3 + \gamma^F(p, q)^{-3}$.

## A.6 EXPERIMENTAL SETUP

In this section, we provide details on the datasets, model configurations, training/finetuning protocols, and evaluation metrics used throughout our experiments. By clarifying each step, we ensure that our methodology is both *transparent* and *reproducible*.

**Dataset.** We evaluate on benchmark datasets such as SST-2, MNLI, RTE, QNLI, QQP, and CoLA to cover diverse classification objectives (sentiment analysis, natural language inference, and question classification). Each dataset is split into training, validation, and test sets following standard protocols (e.g., the GLUE benchmark (Wang, 2018)). We tokenize inputs using the *default* subword tokenizer for each model (e.g., BERT's WordPiece or RoBERTa's Byte-Pair Encoding), lowercasing as necessary. For SALMAN-Guided Attack experiment, we use AdvBench Harmful Behaviors dataset. JailBreak does not involve a training process, thus we did not split the dataset. We directly ranked the entire dataset of 520 data points.

**Language Model.** We evaluate on several benchmark language models such as BERT-base-uncased Devlin (2018), RoBERTa-base Liu (2019), DistilBERT-base-uncased Sanh (2019), ALBERT-base-v2 Lan (2019), GPT-2 Radford et al. (2019), and LLaMA-7B-v2 Touvron et al. (2023).

Table 10: Summary of hyperparameters ($k$ is for kNN graph construction and SPF is for our low-rresistance-diameter decomposition) used in our method for each *(model, attack)* configuration. DIS refers to random selection of deletion, insertion, or swap.

| Model | Attack | $k$ | $SPF$ |
|---|---|---|---|
| *SST-2* | | | |
| BERT-base-uncased | DIS | 30 | 2 |
| RoBERTa-base | DIS | 30 | 2 |
| DistilBERT-base-uncased | DIS | 30 | 2 |
| ALBERT-base-v2 | DIS | 30 | 2 |
| GPT-2 | DIS | 10 | 2 |
| LLaMA-7B-v2 | DIS | 10 | 2 |
| GPT-2 | spaCy | 20 | 2 |
| LLaMA-7B-v2 | spaCy | 30 | 3 |
| GPT-2 | TextAttack | 10 | 2 |
| LLaMA-7B-v2 | TextAttack | 10 | 2 |
| *MNLI* | | | |
| BERT-base-uncased | DIS | 30 | 2 |
| RoBERTa-base | DIS | 30 | 2 |
| DistilBERT-base-uncased | DIS | 30 | 2 |
| ALBERT-base-v2 | DIS | 30 | 2 |
| GPT-2 | DIS | 50 | 2 |
| LLaMA-7B-v2 | DIS | 10 | 2 |
| GPT-2 | spaCy | 70 | 3 |
| LLaMA-7B-v2 | spaCy | 70 | 2 |
| GPT-2 | TextAttack | 20 | 3 |
| LLaMA-7B-v2 | TextAttack | 10 | 2 |

**Hyperparameter settings.** We obtain *Distance Mapping Distortion* scores for each sample by comparing input and output manifold distances (e.g., from $\mathbf{z}_X$ to $\mathbf{z}_Y$). Summary of hyperparameters during DMD calculation is in Table 10. To gauge how much finetuned models deviate from their pretrained checkpoints, we reference layer-wise similarity metrics such as CKA and STIR (Neerudu et al., 2023).

k-NN Ablation on SST-2 (GPT-2). We also evaluate the sensitivity of SALMAN to the choice of $k$ in $k$-NN graph construction. Specifically, we vary $k \in \{15, 20, 30\}$ for GPT-2 on SST-2 and compare the *Kullback–Leibler Divergence* (KLD) and *BERTScore* (Precision, Recall, $F_1$) for non-robust (NR) vs. robust (R) samples. As shown in Table 11, increasing $k$ does not drastically alter the distinction between robust and non-robust data; the non-robust subsets consistently exhibit higher KLD and slightly lower BERTScores, while robust subsets remain more stable under perturbations. This indicates that SALMAN is relatively insensitive to moderate changes in $k$.

Table 11: Effect of varying $k$ in $k$-NN on robustness and similarity metrics (GPT-2, SST-2).

| $k$ (kNN) | KLD (NR) | KLD (R) | Precision (NR) | Recall (NR) | $F_1$ (NR) | Precision (R) | Recall (R) | $F_1$ (R) |
|---|---|---|---|---|---|---|---|---|
| 15 | 0.1110 | 0.0000 | 0.9972 | 0.9978 | 0.9975 | 0.9990 | 0.9988 | 0.9989 |
| 20 | 0.0988 | 0.0003 | 0.9973 | 0.9978 | 0.9976 | 0.9993 | 0.9992 | 0.9992 |
| 30 | 0.1923 | 0.0000 | 0.9962 | 0.9970 | 0.9966 | 0.9992 | 0.9992 | 0.9992 |

## A.7 LAYER-WISE STIR AND CKA RESULTS

In addition to the layer-wise comparison between normal and guided fine-tuning shown in Table 12 (CoLA dataset), we replicate the same analysis for the **SST-2** and **RTE** tasks under GPT-2. Following the exact protocol of Section 4.3 and Neerudu et al. (2023), we assign higher training weights to non-robust data (determined by our DMD ranking) and lower weights to robust data. As before, we measure:

- **Validation Accuracy** on the downstream task,

Table 12: Layer-wise STIR and CKA for GPT-2 on CoLA. Each cell shows the "Normal fine-tuning / SALMAN-guided fine-tuning", rounded to four decimal places. Normal fine-tuning validation accuracy is $0.7468$, SALMAN-guided fine-tuning validation accuracy is $0.7709$. $m1$ is the pre-trained model and $m2$ is the fine-tuned model. Better results are in **bold**.

| Layer | STIR(m2m1) | STIR(m1m2) | CKA |
|-------|-----------|-----------|-----|
| 0 | 0.9623 / **0.9623** | 0.9623 / **0.9623** | 1.0000 / **1.0000** |
| 1 | 0.9070 / **0.9073** | **0.9065** / 0.9053 | 0.9987 / **0.9987** |
| 2 | 0.9688 / **0.9691** | 0.9690 / **0.9711** | **0.9936** / 0.9931 |
| 3 | 0.9848 / **0.9904** | **0.9678** / 0.9551 | **0.9856** / 0.9748 |
| 4 | 0.9853 / **0.9934** | 0.9690 / **0.9775** | 0.9836 / **0.9837** |
| 5 | 0.9904 / **0.9928** | 0.9750 / **0.9752** | 0.9906 / **0.9909** |
| 6 | 0.9897 / **0.9924** | 0.9697 / **0.9767** | 0.9920 / **0.9945** |
| 7 | 0.9895 / **0.9927** | 0.9724 / **0.9833** | **0.9931** / 0.9909 |
| 8 | 0.9860 / **0.9914** | 0.9680 / **0.9854** | 0.9923 / **0.9936** |
| 9 | 0.9825 / **0.9872** | 0.9666 / **0.9770** | 0.9905 / **0.9907** |
| 10 | 0.9776 / **0.9833** | 0.9647 / **0.9762** | 0.9917 / **0.9928** |
| 11 | 0.9730 / **0.9784** | 0.9628 / **0.9678** | 0.9893 / **0.9904** |
| 12 | 0.4691 / **0.7233** | 0.6819 / **0.7924** | 0.5612 / **0.7251** |

- **STIR** (Similar Token Identity Representation) metrics (`m2m1`, `m1m2`) capturing how similar layer $i$ in the fine-tuned model $m2$ is to layer $j$ in the pre-trained model $m1$,
- **CKA** measuring layer-wise alignment between $m1$ and $m2$ embeddings.

**SST-2 Results.** Table 14 shows GPT-2's layer-wise STIR and CKA under normal vs. guided fine-tuning on SST-2. Both approaches yield similar *final accuracy* (0.9231 vs. 0.9232), yet the guided variant consistently achieves higher STIR/CKA scores in later layers. In particular, layer 12 sees a substantial jump in STIR(`m2m1`) from 0.0533 to 0.0867 and CKA from 0.1459 to 0.2039, indicating closer alignment to the pre-trained checkpoint.

**RTE Results.** In Table 13, we compare normal vs. guided fine-tuning for GPT-2 on the RTE dataset. While both runs converge similarly in accuracy (not shown here to save space), the guided approach again shows improved STIR and CKA alignment with the pre-trained checkpoint. For instance, layer 12 sees an increase from 0.2858 to 0.3393 in STIR(`m2m1`) and from 0.3458 to 0.3476 in CKA.

Table 13: Layer-wise STIR and CKA for GPT-2 on **RTE** under Normal vs. Guided fine-tuning. Each cell shows "Normal Fine-tuning / SALMAN-guided Fine-tuning", rounded to four decimal places. Better results in **bold**.

| Layer | STIR(m2m1) | STIR(m1m2) | CKA |
|-------|-----------|-----------|-----|
| 0 | 0.9913 / **0.9913** | 0.9914 / **0.9914** | 1.0000 / **1.0000** |
| 1 | 0.9786 / **0.9791** | 0.9776 / **0.9779** | **0.9986** / 0.9977 |
| 2 | **0.9859** / 0.9857 | **0.9859** / 0.9852 | 0.9976 / **0.9990** |
| 3 | 0.9903 / **0.9920** | 0.9917 / **0.9918** | 0.9951 / **0.9987** |
| 4 | 0.9897 / **0.9902** | 0.9897 / **0.9900** | 0.9885 / **0.9963** |
| 5 | 0.9898 / **0.9908** | 0.9891 / **0.9916** | 0.9894 / **0.9981** |
| 6 | 0.9865 / **0.9872** | 0.9869 / **0.9886** | 0.9908 / **0.9923** |
| 7 | 0.9821 / **0.9829** | 0.9806 / **0.9839** | 0.9746 / **0.9801** |
| 8 | 0.9758 / **0.9781** | 0.9709 / **0.9761** | 0.9407 / **0.9500** |
| 9 | 0.9708 / **0.9724** | 0.9607 / **0.9696** | 0.9288 / **0.9492** |
| 10 | 0.9564 / **0.9601** | 0.9359 / **0.9507** | 0.9028 / **0.9347** |
| 11 | **0.9333** / 0.9331 | 0.9152 / **0.9265** | 0.9223 / **0.9390** |
| 12 | 0.2858 / **0.3393** | 0.6131 / **0.6203** | 0.3458 / **0.3476** |

**Discussion.** Similar to our observations on CoLA (Table 12), placing higher emphasis on non-robust data (i.e., higher DMD samples) preserves downstream performance while bringing the fine-tuned layers closer to the original pre-trained representations. These improvements in STIR and

Table 14: Layer-wise STIR and CKA for GPT-2 on **SST-2** under Normal vs. Guided fine-tuning. Each cell shows "Normal Fine-tuning / Guided Fine-tuning", rounded to four decimal places. **Acc** is the validation accuracy of each method. For STIR, (m2m1) compares the fine-tuned model $m2$ to the pre-trained model $m1$, and (m1m2) is the reverse; CKA measures embedding similarity. Better results are in **bold**.

| Validation Accuracy: Normal = 0.9231, SALMAN-guided = 0.9232 | | | |
|---|---|---|---|
| **Layer** | **STIR(m2m1)** | **STIR(m1m2)** | **CKA** |
| 0 | 0.9913 / **0.9913** | 0.9912 / **0.9912** | 1.0000 / **1.0000** |
| 1 | 0.9763 / **0.9771** | 0.9784 / **0.9787** | 0.9963 / **0.9974** |
| 2 | 0.9784 / **0.9789** | 0.9762 / **0.9767** | 0.9971 / **0.9973** |
| 3 | 0.9703 / **0.9713** | 0.9366 / **0.9414** | 0.9460 / **0.9469** |
| 4 | 0.9661 / **0.9715** | 0.9549 / **0.9608** | 0.9773 / **0.9800** |
| 5 | 0.9736 / **0.9757** | 0.9469 / **0.9589** | 0.9705 / **0.9738** |
| 6 | 0.9649 / **0.9704** | 0.9343 / **0.9450** | 0.9568 / **0.9604** |
| 7 | 0.9618 / **0.9672** | 0.9389 / **0.9476** | 0.9642 / **0.9675** |
| 8 | 0.9663 / **0.9703** | 0.9514 / **0.9598** | 0.9800 / **0.9825** |
| 9 | 0.9435 / **0.9553** | 0.9473 / **0.9545** | 0.9717 / **0.9787** |
| 10 | 0.9230 / **0.9504** | 0.9486 / **0.9573** | 0.9599 / **0.9774** |
| 11 | 0.8567 / **0.9208** | 0.9328 / **0.9426** | 0.9166 / **0.9562** |
| 12 | 0.0533 / **0.0867** | 0.7504 / **0.7755** | 0.1459 / **0.2039** |

CKA suggest *reduced representational drift*, consistent with the intuition that focusing on "hard" samples forces the model to retain more generalizable features from pre-training (Cheng et al., 2021; Zhu et al., 2023a).

Overall, these extended results on SST-2 and RTE corroborate our main findings: *robustness-guided fine-tuning* effectively balances task performance with better alignment to the pre-trained checkpoint across multiple datasets.

## A.8    WEIGHTED FINE-TUNING AND INTEGRATION WITH ROSE

**Motivation: Focus on Non-Robust Data.**    As discussed in Section 4.3, prior studies have shown that directing more attention to non-robust ("hard") samples during training can improve model generalizability and resilience (Cheng et al., 2021; Zhu et al., 2023a). Our approach identifies these difficult samples via the SALMAN-based ranking and then *assigns higher training weights* to them, while simultaneously down-weighting samples that appear robust. We follow the finetuning protocol of Neerudu et al. (2023), hypothesizing that emphasizing harder samples preserves more of the pre-trained model's versatility. This reduces the risk of overfitting to "easy" data and yields representations closer to the original checkpoint (see STIR/CKA results in Appendix A.7).

### A.8.1    WEIGHTING SCHEMES FOR GUIDED FINE-TUNING

**Linear Schedule.**    We sort all training samples in descending order of their DMD values (highest DMD = most non-robust), then map each sample to a weight $w \in [0, 1]$ proportional to its position in this ranking. Concretely, if the highest-DMD sample is indexed as rank 0, it receives weight $\approx 1.0$, whereas the lowest-DMD sample (rank $n-1$) receives weight near 0.0. Intermediate samples smoothly interpolate between these extremes.

**Combining with SOTA Robust Training (ROSE).**    We further integrate our DMD-based weighting into **ROSE: Robust Selective Fine-tuning** (Jiang et al., 2022), which filters out spurious parameter updates by comparing dropout-induced distributions at each iteration:

$$L_{\text{KL}}^{(t)} \;=\; D_{\text{KL}}(P_t \parallel P_t') \;+\; D_{\text{KL}}(P_t' \parallel P_t).$$

ROSE removes parameter changes that inflate $L_{\text{KL}}$ excessively, thus improving adversarial resilience.

**Per-sample Weight $w(x)$ for Joint Optimization**    Alternatively, we employ a logistic transition-based function partitioned into intervals:

- **Top-25% Non-Robust** (i.e., highest DMD) can receive a *larger* weight (e.g., 2.0),

- **Middle-Range** samples gradually decrease from 1.0 to 0.0 in stepwise logistic transitions,

- **Bottom-5% Most Robust** eventually gets weight 0.0 (or near-zero).

This piecewise approach allows a *finer distinction* between very hard, moderately challenging, and trivially easy samples.

**Joint Optimization.** We incorporate our per-sample weight $w(x)$ into ROSE's fine-tuning loss. Specifically, if the original ROSE objective is

$$\mathcal{L}_{\text{ROSE}}(\theta_t) \;=\; \mathbb{E}_{x \sim \mathcal{D}}\big[L_{\text{task}}(x, \theta_t) + \lambda L_{\text{KL}}^{(t)}\big],$$

then our *combined* objective is

$$\mathcal{L}_{\text{ROSE+Guided}}(\theta_t) \;=\; \mathbb{E}_{x \sim \mathcal{D}}\Big[w(x) \cdot L_{\text{task}}(x, \theta_t) + \lambda L_{\text{KL}}^{(t)}\Big].$$

Hence, the model is "selective" not only at the *parameter* level (via $L_{\text{KL}}$) but also at the *sample* level (via DMD-based weighting).

A.9 FROM PGM TO MANIFOLD: VALIDATING ON GRAPH BENCHMARKS

Although our primary interest is applying the PGM-based manifold to NLP data (where nodes represent text embeddings), we first validate how well our spectral sparsification and resistance distance preservation works on *standard graph benchmarks*, namely **Cora**, **Citeseer**, and **PolBlogs**. These datasets are widely used in the GNN literature and offer:

- **Well-defined adjacency**: Each graph provides a clear baseline for measuring changes in effective resistance.

- **Known benchmarks for graph-based algorithms**: This allows direct comparison of spectral or manifold-like approaches without the additional complexity of NLP text embedding.

In other words, while our ultimate goal is to build a *manifold* for robustness analysis in transformer-based language models, these classic graph datasets serve as an *intermediary check* to confirm that the PGM manifold indeed preserves *resistance distances* in large-scale graphs.

**Why Graph Benchmarks Instead of NLP Data?**

- *Ground-Truth Adjacency*: For cora/citeseer/polblogs, the adjacency matrix is explicitly available, enabling a direct before/after comparison of edge sparsity and distance correlation. In contrast, NLP data initially lacks a clear "graph," so we must approximate edges (e.g., via $k$-NN). Verifying the correctness of our approach on well-studied graph datasets ensures that the spectral sparsification steps *properly* preserve distances.

- *Easier Resistance Verification*: By default, each node in these graph benchmarks is associated with a known set of neighbors. We can compute full-pairwise effective resistance or measure Pearson, Spearman, and MSE between original and sparsified graphs (Table 15). This level of straightforward measurement is less trivial in NLP tasks, where adjacency depends on embedding similarity.

**Experiment Setup.**

1. **Compute original resistance distances** for each pair of nodes in the unsparsified graph.

2. **Apply our SPF (Spectral Pruning via effective-resistance)** procedure at various parameters (e.g., `param` $\in \{2, 3, 4\}$), generating a pruned graph that discards edges with smaller distance ratios.

3. **Quantify distance preservation** via Pearson correlation, Spearman correlation, MSE, and relative error (`RelErr`) between the original and the pruned graph's resistance distances.

4. **Measure final edge count** as a fraction of the original adjacency size.

Table 15: SPF results on three datasets (Cora, Citeseer, Polblogs). For each dataset, we vary the SPF parameter in $\{2, 3, 4\}$, then measure how well the transformed adjacency preserves the original resistance distances (Pearson / Spearman correlation, MSE, relative error). "Edges%" indicates the proportion of edges retained relative to the original graph.

| Dataset | SPF | Pearson | Spearman | MSE | RelErr | Edges% |
|---|---|---|---|---|---|---|
| cora | 2 | 0.9029 | 0.8899 | 0.58045 | 0.3511 | 80.29% |
| cora | 3 | 0.8602 | 0.8495 | 1.01185 | 0.5178 | 74.51% |
| cora | 4 | 0.8113 | 0.7988 | 1.76080 | 0.7074 | 70.21% |
| citeseer | 2 | 0.9475 | 0.9475 | 0.89848 | 0.2463 | 80.48% |
| citeseer | 3 | 0.9220 | 0.9190 | 1.67925 | 0.3658 | 75.71% |
| citeseer | 4 | 0.9074 | 0.9014 | 2.46463 | 0.4674 | 72.25% |
| polblogs | 2 | 0.9565 | 0.9693 | 0.02916 | 0.3209 | 67.58% |
| polblogs | 3 | 0.9090 | 0.9356 | 0.07819 | 0.6778 | 53.19% |
| polblogs | 4 | 0.8323 | 0.8696 | 0.20026 | 1.4342 | 37.03% |

**Results and Analysis.** Table 15 summarizes the outcomes on **Cora**, **Citeseer**, and **PolBlogs**. For each dataset:

- **Pearson & Spearman correlation** remain high ($> 0.80$) even when we prune roughly 20-40% of the edges, confirming that the principal global and local distance structures remain intact.
- **MSE and RelErr** naturally increase with more aggressive pruning, yet remain within acceptable ranges for many use-cases (e.g., GNN training, manifold-based clustering).
- **Sparsification Rate** (`Edges%`) indicates that by increasing the SPF parameter, we can achieve increasingly compact graphs without catastrophically degrading the resistance-distance correlation.

In short, these results validate that our spectral-pruning approach effectively maintains key *manifold properties* (represented by resistance distances) across standard graph benchmarks. By extension, we expect similar fidelity in large-scale NLP tasks once we construct an initial $k$-NN or adjacency graph from text embeddings.

Having verified the correctness of our PGM manifold construction on well-known graph datasets, we now apply the same principles (near-linear spectral sparsification plus Laplacian-based $\Theta$ construction) to build manifolds for high-dimensional text embeddings. This ensures that the subsequent distance analyses in our transformer robustness framework rely on an *accurate* and *scalable* manifold, preserving essential local and global distances just as effectively as in these classic graph scenarios.

## A.10 EFFECTIVE RESISTANCE DISTANCE

**Motivation and Intuition.** In graph-based methods, the *effective resistance distance* (also called *resistance distance* in electrical-network parlance) provides a powerful metric for understanding the relationship between pairs of nodes. Unlike simple shortest-path lengths, effective resistance captures both local and global connectivity: if two nodes are connected by many parallel paths, they have lower effective resistance than nodes primarily joined by a single, bottleneck path (Spielman & Teng, 2011).

**Electrical Network Interpretation.** One way to grasp effective resistance is to imagine placing a 1-Ohm resistor on each edge of the graph and then viewing the entire graph as an electrical circuit:

- Inject 1 amp of current into node $u$ and extract it from node $v$.
- Let $\varphi(x)$ be the resulting electrical potential at any node $x$ in the network.
- The *effective resistance distance* $R_{\text{eff}}(u, v)$ is then *the voltage difference* between $u$ and $v$, i.e., $\varphi(u) - \varphi(v)$, required to sustain that 1-amp current.

Thus, if there are many alternative routes (parallel edges) from $u$ to $v$, the network offers "lower resistance" between them, indicating $u$ and $v$ are closely tied in the graph's connectivity structure (Chandra et al., 1989; Ellens et al., 2011).

**Mathematical Formulation via Laplacian Pseudoinverse.** Let $G = (V, E)$ be an undirected, connected graph with $n = |V|$ nodes. Denote its *Laplacian matrix* by $L_G = D - W$, where $D$ is the diagonal degree matrix and $W$ is the adjacency (or edge-weight) matrix. Since $L_G$ is positive semidefinite and has rank $n-1$ for a connected graph, it admits a Moore-Penrose pseudoinverse $L_G^+$ (Mohar, 2004; Spielman & Teng, 2011). For nodes $p$ and $q$:

$$R_{\text{eff}}(p, q) = (e_p - e_q)^\top L_G^+ (e_p - e_q),$$

where $e_p$ is the standard basis vector (all zeros except a 1 in the $p$-th coordinate). Intuitively, $L_G^+$ encodes global connectivity, so $R_{\text{eff}}(p, q)$ measures "how difficult it is to flow current" from $p$ to $q$ across $G$ (Babić et al., 2002).

**Example: Line Graph vs. Square Graph.** To illustrate how the *effective resistance* distance can differ substantially from the naive (hop-count) distance, consider:

- **Line Graph with 3 Nodes** $\{1, 2, 3\}$ and unit-weight edges $(1, 2)$ and $(2, 3)$. The hop distance from node 1 to node 3 is 2. When modeled as a resistor network, each edge contributes 1 ohm in series; thus, the effective resistance between node 1 and node 3 is

$$R_{\text{eff}}(1, 3) = 1 + 1 = 2.$$

- **Square Graph with 4 Nodes** $\{1, 2, 3, 4\}$ and edges $(1, 2)$, $(2, 3)$, $(3, 4)$, $(4, 1)$, each of unit weight. The naive (hop) distance from node 1 to node 3 is 2 (e.g., via $1 \to 2 \to 3$ or $1 \to 4 \to 3$). However, in the resistor-network view, there are two distinct 2-edge paths running in *parallel* between node 1 and node 3:

$$1 \to 2 \to 3 \quad \text{and} \quad 1 \to 4 \to 3.$$

Each path alone would have resistance $1 + 1 = 2$. Because they are in parallel, the total effective resistance is

$$R_{\text{eff}}(1, 3) = \left(\frac{1}{2} + \frac{1}{2}\right)^{-1} = 1.$$

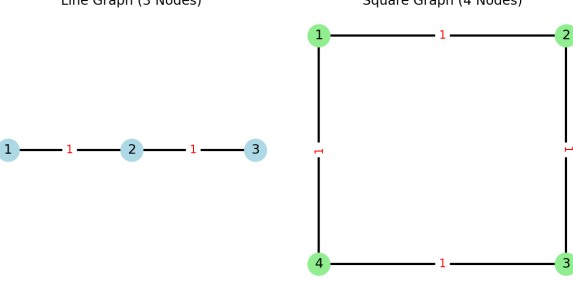

Figure 3: **Line vs. Square Graph Examples.** *(Left)* The line graph with nodes $\{1, 2, 3\}$. *(Right)* The square graph with nodes $\{1, 2, 3, 4\}$. Although both have a hop distance of 2 between node 1 and node 3, the *effective resistance* differs significantly: it is $R_{\text{eff}}(1, 3) = 2$ in the line graph (two edges in series), versus $R_{\text{eff}}(1, 3) = 1$ in the square graph (two parallel 2-edge paths).

These simple examples illustrate that the effective resistance distance may diverge from the naive, purely local distance. For node pairs in a graph with parallel paths, the effective resistance is often smaller than the hop count would suggest. By contrast, if all paths between two nodes lie strictly in series (as in a line graph), the effective resistance grows as a sum of edge resistances. Such distinctions are at the heart of why resistance-based metrics can better capture global connectivity and structural nuances in graph-based manifold analysis.

**Relevance to Robustness and Graph Learning.** The notion of effective resistance has become increasingly relevant for:

- **Spectral Graph Sparsification**: Low- and high-resistance edges are treated differently; small-resistance edges indicate redundancy, enabling fast approximation algorithms (Spielman & Srivastava, 2011; Spielman & Teng, 2011).

- **Commute/Random Walk Times**: $R_{\text{eff}}(p, q)$ also relates to the expected commute time of a random walk between $p$ and $q$ (Chandra et al., 1989), linking local connectivity to global diffusion properties.

- **Manifold Preserving Embeddings**: By preserving effective resistance distances, one can maintain both local neighborhoods and global circuit-like structure in a final embedding or graph model (Ellens et al., 2011; Feng, 2021).

In short, effective resistance unifies local and global connectivity aspects, making it ideal for measuring how perturbations might propagate through a network—and by extension, how to keep the manifold structure stable in large-scale data (e.g., NLP embeddings).

## A.11 EMPIRICAL EVIDENCE OF $(\gamma_{\min}^F)^{-1}$ CAPTURING "COLLAPSES"

In Section 3.3, we highlighted how a large $(\gamma_{\min}^F)^{-1}$ indicates another dimension of fragility: *distant* inputs becoming overly close in the output space. Below, we provide empirical results on multiple model–task combinations, measuring:

- **Cosine Similarity (Cos)** between original vs. perturbed embeddings,

- **KL Divergence (KLD)** between output distributions,

- for both *non-robust* vs. *robust* samples, under A: $\gamma_{\max}^F$ or B: $\gamma_{\max}^F + (\gamma_{\min}^F)^{-1}$ setting.

A significant gap in Cos or KLD between robust and non-robust samples suggests the model *amplifies* small differences in the non-robust subset (or "collapses" large differences). Conversely, if robust samples remain stable, it aligns with a lower distortion (or higher $\gamma_{\min}^F$).

Table 16: Comparisons of Cosine Similarity (Cos) and KL Divergence (KLD) across *non-robust* vs. *robust* subsets, under **A:** $\gamma_{\max}^F$ **or B:** $\gamma_{\max}^F + (\gamma_{\min}^F)^{-1}$ setting. Selected samples are attacked by spaCy. Each row shows: (1) model+dataset, (2)(3) Non-robust Cos, (4)(5) Robust Cos, (6)(7) Non-robust KLD, (8)(9) Robust KLD. Higher Cos / lower KLD typically indicates more stable behavior. Better results are in **bold**.

| Model + Task | Non-rob Cos | | Rob Cos | | Non-rob KLD | | Rob KLD | |
|---|---|---|---|---|---|---|---|---|
| | A | B | A | B | A | B | A | B |
| **BERT, RTE** | 0.9194 | **0.9091** | 0.9282 | **0.9407** | 0.00794 | **0.00884** | 0.00709 | **0.00605** |
| **BERT, SST-2** | 0.9368 | **0.9358** | 0.9968 | **0.9969** | 0.00631 | **0.00647** | 0.00033 | **0.00032** |
| **GPT-2, RTE** | 0.9755 | **0.9662** | 0.9844 | **0.9917** | 0.01992 | **0.01992** | 1.14e-13 | **9.44e-14** |
| **GPT-2, SST-2** | 0.9730 | **0.9634** | 0.9989 | **0.9988** | 0.12331 | **0.15453** | 2.21e-06 | **4.38e-07** |
| **LLaMA-7Bv2, RTE** | 0.9511 | **0.9438** | 0.9537 | **0.9582** | 0.6998 | **0.7733** | 0.6764 | **0.4797** |
| **LLaMA-7Bv2, SST-2** | **0.9490** | 0.9491 | 0.9777 | **0.9779** | **0.53032** | 0.52974 | 0.21646 | **0.17981** |

**Observations.**

Empirically, when we *combine* both $\gamma_{\max}^F$ and $(\gamma_{\min}^F)^{-1}$ (e.g., by ranking samples via $\gamma_{\max}^F + (\gamma_{\min}^F)^{-1}$), we obtain a more accurate partition of robust vs. non-robust data than using $\gamma_{\max}^F$ alone. Specifically:

- **Robust subset** selected by $\left[\gamma_{\max}^F + (\gamma_{\min}^F)^{-1}\right]$ displays *higher* cosine similarity and *lower* KLD relative to a purely $\gamma_{\max}^F$-based choice,

- **Non-robust subset** exhibits *lower* cosine similarity and *higher* KLD, indicating stronger local instability.

This confirms that jointly considering *expansions* ($\gamma_{\max}^F$) and *collapses* ($(\gamma_{\min}^F)^{-1}$) provides a more fine-grained characterization of model robustness—reinforcing the notion that both extremes of the distortion spectrum matter for local manifold analysis.

## A.12 SCALABILITY AND EFFICIENCY

Table 17 reports total wall-clock time (in seconds) for *embedding the dataset*, *constructing the manifold graph*, and *computing DMD* on standard hardware. Notably, even the largest GLUE tasks remain tractable. For instance, MNLI (393k samples) takes $\approx 6060$ seconds ($\sim 1.7$ hours), which is a one-time cost. Smaller tasks like QNLI (105k) finish in $\sim 12$ minutes. These results underscore that SALMAN is viable for mainstream NLP benchmarks. For extremely large datasets, approximate or distributed strategies can be employed for further scalability.

Table 17: SALMAN runtime across different GLUE tasks. Approx. sample counts and total runtime on typical hardware.

| Dataset | #Samples | Runtime (sec) |
|---------|----------|---------------|
| SST-2 | $\sim 67$k | 642.4 |
| RTE | $\sim 2.5$k | 12.0 |
| QNLI | $\sim 105$k | 736.3 |
| MNLI | $\sim 393$k | 6060.2 |

Thus, while SALMAN does require a modest upfront cost to build the manifold and compute distortions, the resulting robustness ranking can be reused for downstream tasks (e.g., adversarial evaluation, fine-tuning). This amortizes the cost and keeps the approach practical for modern NLP pipelines.

## A.13 MORE ATTACK EXPERIMENT RESULTS

Figure 4 (left) shows that ASR is highest for the first decile (most non-robust) and consistently decreases as samples become more robust in higher deciles. This confirms that SALMAN ranking provides a reliable gradient for identifying vulnerable data points.

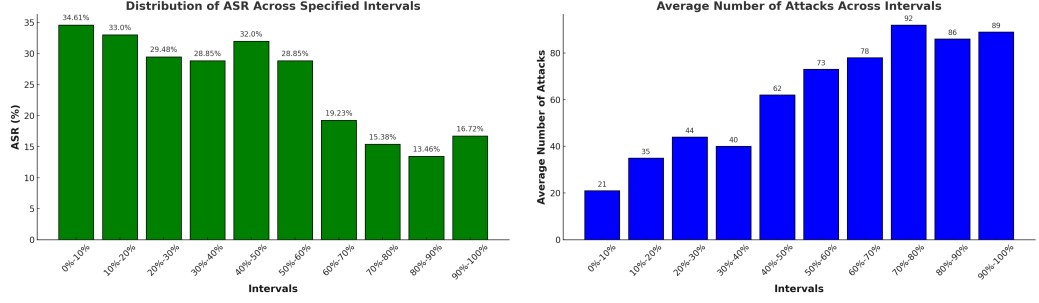

Figure 4: *(Left)* Attack Success Rate (ASR) across deciles of non-robustness. *(Right)* Average number of AutoDAN steps needed for successful attack on robust vs. non-robust subsets. Non-robust samples require fewer steps, highlighting their vulnerability.

### A.13.1 MEASURING ATTACK STEPS

We further follow GCG Zou et al. (2023) and AutoDAN Liu et al. (2023) to measure the average number of attack steps required. By default, GCG uses a fixed 250 steps for each trial, but we adapt the AutoDAN approach to run up to 100 steps. Figure 4 (right) shows that non-robust samples require *significantly fewer* steps for successful attack, whereas robust samples demand more queries to break. This corroborates our SALMAN-based ranking.

### A.13.2 SALMAN SCORE FROM DIFFERENT LAYERS

To empirically examine the sensitivity of SALMAN scores to layer selection, we conducted additional experiments using embeddings from intermediate layers on the attacking task. Specifically, we evaluated two configurations:

- Setup 1: Raw input embeddings as the input manifold and embeddings from the 16th-layer MHSA as the output manifold.
- Setup 2: Embeddings from the 16th-layer MHSA as the input manifold and embeddings from the final layer as the output manifold.

For both setups, we ranked the top 1% most non-robust samples and evaluated their robustness under adversarial attacks using the AutoDAN framework. The results show that Setup 1 achieved an attack success rate of 20% (1 successful attack out of 32 attempts), while Setup 2 yielded a comparable 20% (1 successful attack out of 40 attempts). In contrast, our original design, which utilized embeddings from the initial and final layers, achieved substantially higher attack success rates with fewer attempts. These findings indicate that intermediate-layer embeddings provide less effective robustness ranking, thereby validating the soundness of our original methodological choice.

### A.13.3 PROXY-BASED SALMAN RANKING

One may wonder if SALMAN must be computed on the exact same model we later attack. We investigate using GPT-2, LLaMA2-7B, or LLaMA3-8B embeddings as a "proxy" for SALMAN ranking, then testing the transferability of the attack to the target LLM. Table 5 shows that the Attack Success Rate (ASR) remains quite similar across each proxy's ranking, suggesting that SALMAN is fairly robust to model variations.

### A.13.4 ATTACK ON MULTILINGUAL

The last attack experiment investigates the multilingual setting. We conducted evaluations on the Chinese subset of the MultiJail dataset, which contains 316 samples Deng et al.. Using embeddings from LLaMA-8B, SALMAN was applied to rank samples by robustness, and adversarial attacks were subsequently carried out on GPT-4o. Without ranking, the overall attack success rate across the full dataset was 18.4%. In contrast, focusing on the top 10% most non-robust samples identified by SALMAN yielded a substantially higher success rate of 37.5% (12 out of 32). These findings highlight the initial effectiveness of SALMAN in multilingual contexts and underscore its potential applicability for broader cross-lingual adversarial evaluation.

