# OpenReview forum: "SALMAN: Stability Analysis of Language Models Through the Maps Between Graph-based Manifolds"
_ICLR.cc/2026/Conference — ICLR 2026 Conference Withdrawn Submission_

### Official Review · Reviewer_vMVJ · 2025-10-29

**Soundness:** 3
**Presentation:** 2
**Contribution:** 3
**Rating:** 4
**Confidence:** 3

**Summary:**

The paper introduces a novel way to assess the robustness of an individual sample. To achieve this, the authors estimate the geometry of the input space and output space and compare the distance samples move in the input space versus the output space. Based on this idea, they derive a score they call the SALMAN score, which estimates how a sample p moves with respect to samples in its neighborhood. To compute this score efficiently, the authors propose a novel approach to estimate a precision matrix that captures local connectivity. Next, they show that their score outperforms simple baselines, can be used to guide adversarial attacks, and achieves more efficient adversarial fine-tuning.

**Strengths:**

- The problem statement is clearly motivated.
- The claims are mostly supported theoretically and with experiments.
- The new method for optimizing the precision matrix is both elegant and well justified
- Connecting input and output geometry using a graph-derived geometry to estimate sample-specific robustness is novel and, given the results, promising.
- Showing that the geometry of the input space is a valuable contribution.

**Weaknesses:**

Weaknesses

- The paper would benefit greatly from a dedicated preliminaries section that introduces the necessary concepts, such as PGMs and the metrics used in the paragraph “Effective-resistance distance.” Additionally, notation such as e_{p,q} is not
introduced.


- The objective for the PGM is concave and can be solved efficiently with projected gradient descent. It would be important to evaluate how well your algorithm approximates the optimal solution. This can also happen in a smaller setting where the typical O(N^3) algorithm still terminates in a reasonable time.

- It is not clear how using the embeddings allows you to make statements about the input–output manifold. The embeddings after MHSA are not really inputs anymore. This requires more explanation and/or proofs.

- Reporting the cosine similarity makes sense, yet it underballs the difference between the embeddings. The arccos is not linear in that domain. To give an example, arccos(0.9772)$\approx$12.26° and
arccos(0.9981)=3.53°, which shows quite a difference, while the cosine similarities that you report do not. This would both emphasize your contribution and simplify the interpretation of the results.

- In my opinion, the section: Salman-Guided attack, takes the wrong storyline. In an attack scenario, you cannot necessarily choose the sample you want to attack. This section would be better positioned as verifying that your ranking correlates strongly with how easy it is to attack a sample.

- Another simple baseline would be to estimate the local curvature in embedding space by sampling in an epsilon ball around the embedding and assessing how the output changes. This can be very efficiently computed with appropriate caching and batched inference.

Minor:
- The Theorems are too informal; it would be better to have more detailed statements to know exactly what is proven.

- It would be better to first introduce the intuition of your SALMAN score and then show the formulas.

Note: The paper introduces an interesting novel method that is theoretically well-founded and shows promising results. Yet, it does not properly present these and lacks the necessary details to understand the method and experiments in full detail. Thus, I am leaning toward a reject, but with appropriate improvements, clarifications, and extra baselines, I am inclined to raise my score without making any promises to do so.

**Questions:**

How does your approach solve the problem of the discrete nature of token embeddings if you still use embeddings?

How exactly do you generate the PGM? And what is probabilistic here?

Can you comment on how your approach relates to estimating the local curvature around a sample?

---

> ### Author Response · Authors · 2025-11-21
>
> We sincerely thank all reviewers for the constructive and insightful feedback. Below are the responses to the reviewer's comments and questions.
>
> **W1** introduces the necessary concepts
>
> Thank you for pointing this out. **Due to page limits**, we placed several background items in the appendix, e.g., **Effective‑resistance distance (Appendix A10)**. And we see how that makes the main text harder to parse.  For a graph with Laplacian (L), the **effective resistance** between nodes (p) and (q) is
>
> $R_L(p,q)=e_{p,q}^{\top}L^{+}e_{p,q},$
>
> where $L^{+}$ is the Moore–Penrose pseudoinverse and $e_{p,q}$ is the **incidence vector** with a +1 at index (p), a -1 at index (q), and zeros elsewhere. (Full background and solver details are in **Appendix A10**. We mentioned this again in the revision paper.
>
> **W2** The objective for the PGM is concave and can be solved efficiently with projected gradient descent.
>
> We agree the objective
>
> $
> F(\Theta)=\log\det(\Theta)-\tfrac{1}{k}\operatorname{Tr}(X^\top \Theta X),\qquad
> \Theta=\mathcal L+\sigma^{-2}I
> $
>
> is **concave** in $\Theta$ over the convex set of valid Laplacians (shifted by $\sigma^{-2}I$), and that one can seek the global maximizer via projected/parameterized gradient **ascent** on $\Theta$ (or on edge weights). Our design choice was to favor **near‑linear scalability** while preserving the geometry that our stability metric uses. Specifically, our manifold step relies on the following (quoted verbatim for clarity):
>
> > Theorem 3.1. Maximizing the objective in Equation (1) can be done via an edge pruning strategy equivalent to spectral sparsification of the initial (dense) graph. Edges with small $\rho_{p,q}$ are removed, preserving the essential spectral (and thus resistance) properties of the original graph. The proof is available in Appendix A.2.
> >
>
> This means our learned ($\widetilde{\mathcal L}$) is a $(1\pm\varepsilon)$ **spectral approximation** of the optimizer’s Laplacian $\mathcal L^\star$. Because SALMAN’s distances and distortion ratios use **effective resistances** (which are spectrally stable), this choice targets the geometry most relevant to our metric while avoiding the cubic costs of dense factorizations.
>
> **How close is our solution to the optimum? (theoretical lens)**
>
> If $(1-\varepsilon) \mathcal{L}^\star \preceq \widetilde{\mathcal{L}} \preceq (1+\varepsilon) \mathcal{L}^\star$, then with $\Theta^\star = \mathcal{L}^\star + \sigma^{-2}I$ and $\widetilde{\Theta} = \widetilde{\mathcal{L}} + \sigma^{-2}I$ one also has
> $$
> (1-\varepsilon) \Theta^\star \preceq \widetilde{\Theta} \preceq (1+\varepsilon) \Theta^\star.
> $$
>
> Consequently,
>
> - **Log-det gap.** $n\log(1-\varepsilon) \le \log\det\widetilde{\Theta} - \log\det\Theta^\star \le n\log(1+\varepsilon)$ (dimension $n=|V|$).
> - **Trace term gap.** With $S := \frac{1}{k}XX^\top \succeq 0$,
> $$
> \left|\operatorname{Tr}\big(S(\widetilde{\Theta}-\Theta^\star)\big)\right| \le \varepsilon \operatorname{Tr}(S\Theta^\star).
> $$
>
> Putting these together, the objective deviation satisfies
> $$
> \left|F(\widetilde{\Theta})-F(\Theta^\star)\right| \le n\log\frac{1+\varepsilon}{1-\varepsilon} + \varepsilon \operatorname{Tr}(S\Theta^\star),
> $$
> i.e., an $O(\varepsilon)$ shift whose constants are explicit. Since effective resistances are preserved within $1 \pm \varepsilon$ as well, the DMD ratios and the ensuing SALMAN ranking are correspondingly stable.
>
> **Why not just use projected gradient ascent (PGA) in practice?**
>
> PGA is attractive for **small graphs**, but at the scales we target it is impractical:
>
> - The gradient is $(\nabla_\Theta F=\Theta^{-1}-S)$. Obtaining $(\Theta^{-1})$ (or even a good approximation to all required quadratic forms) typically requires a **Cholesky factorization** or many linear solves per iteration—**(O(n^3))** or at least super‑linear in practice.
> - Projection onto the **Laplacian cone** (PSD, zero row sums, non‑positive off‑diagonals) is itself non‑trivial; a safe route is to **parameterize** $(\mathcal L)$ by non‑negative edge weights (w) via $(\mathcal L=B^\top \mathrm{diag}(w),B)$ (incidence matrix (B)), but then each gradient step needs all edgewise quadratic forms $(b_e^\top \Theta^{-1} b_e)$, again demanding many solves.
>
> Our spectral‑sparsification route (Theorem 3.1) avoids these costs while giving **provable** preservation of the quantities our metric uses.

---

> ### Author Response · Authors · 2025-11-21
>
> **W3 how using the embeddings allows you to make statements about the input–output manifold.**
>
> Our framework does not claim that MHSA embeddings are "raw inputs". We treats them as intermediate representations that capture the model's transformation of the input space. Therefore, the “input manifold” is not the raw token space. It is by definition the set of first‑layer MHSA pooled vectors, $M_X:=E(X)$, and the “output manifold” is the set of last‑layer pooled vectors, $M_Y:=G(E(X))$. We study the internal map $G$ that the rest of the network applies to $E(x)$. Using MHSA as our intermediate representation, therefore, does not change the input manifold we analyze—it is the manifold (the native representation that the subsequent layers already “see”). We will rephrase “input manifold” to “input‑side representation manifold” to make this explicit.
>
> The key insight is that we construct two manifolds:
>
> - **Input manifold (G_X)**: Built from early-layer MHSA outputs (closer to input processing)
> - **Output manifold (G_Y)**: Built from final-layer MHSA outputs (closer to output generation)
>
> As shown in Section 3.1 and Appendix A.1, the MHSA embeddings maintain a **deterministic mapping** from the original token inputs when we fix random seeds and disable dropout. Formally, for input tokens $x$, the transformation through layer l can be expressed as:
>
> $
> z_l = f_l(z_{l-1}) where z_0 = TokenEmbed(x)
> $
>
> This ensures that the manifold structure preserves the input information while capturing how the model transforms it.
>
> Then, the Distance Mapping Distortion (DMD) metric (Eq. 2) measures how the model distorts distances between samples as they flow through the network. This is valid because:
>
> - The effective resistance distance on both manifolds captures semantic similarity
> - The ratio $γ^F(p,q) = d_Y(p,q)/d_X(p,q)$ quantifies how much the model amplifies or collapses distances
> - This holds regardless of whether we use raw tokens or their deterministic transformations
>
> Our experiments (Tables 1-4) demonstrate that samples identified as "non-robust" using this framework indeed show greater sensitivity to input perturbations, validating that our manifold construction captures meaningful model behavior.
>
> **W4 yet it underballs the difference between the embeddings.**
>
> Converting cosine similarities to angular distances reveals more intuitive differences between robust and non-robust samples. For BERT-base on MNLI, the angular distance increased from 14.21° (robust) to 21.13° (non-robust), representing a 48.7% increase. RoBERTa-base showed stability with angular distances below 4° for both sample types. Most notably, GPT-2 demonstrated substantial separation: on MNLI with spaCy attack, robust samples deviated only 2.50° while non-robust samples shifted 9.13°—a 3.65× difference. These angular measurements provide clearer evidence that SALMAN effectively identifies adversarially vulnerable samples, with non-robust samples showing significantly larger embedding perturbations across all models and datasets.
>
> Table 1:
>
> | Model | Dataset | Robust (Mean ± SE)° | Non-Robust (Mean ± SE)° |
> | --- | --- | --- | --- |
> | BERT-base | SST-2 | 21.18° ± 0.67° | 23.38° ± 0.53° |
> | BERT-base | MNLI | 14.21° ± 0.40° | 21.13° ± 0.60° |
> | RoBERTa-base | SST-2 | 1.47° ± 0.06° | 2.43° ± 0.06° |
> | RoBERTa-base | MNLI | 2.57° ± 0.29° | 3.53° ± 0.34° |
>
> Table 2:
>
> | Dataset | Attack | Robust (Mean ± SE)° | Non-Robust (Mean ± SE)° |
> | --- | --- | --- | --- |
> | SST-2 | spaCy | 1.41° ± 0.11° | 6.42° ± 0.24° |
> | SST-2 | TextAttack | 1.99° ± 0.14° | 7.49° ± 0.28° |
> | MNLI | spaCy | 2.50° ± 0.16° | 9.13° ± 0.32° |
> | MNLI | TextAttack | 2.87° ± 0.17° | 8.06° ± 0.30° |
>
> **W5 Attack scenario issue**
>
> In Sec. 4.2, our primary research goal was indeed to **verify that the SALMAN ranking correlates with per‑sample attack difficulty**, using jailbreak methods as the test pipeline. Figure 2 on page 8 and the decile analysis in Appendix A.13 (Figure 4) show that: (i) the attack success rate is highest in the top SALMAN deciles and decreases monotonically as we move to more “robust” bins, and (ii) the average number of attack steps required increases in the same direction, across both GCG and AutoDAN and multiple LLMs. Table 5 further shows that this holds even when the ranking is computed on a proxy model and transferred to another target LLM.
>
> While we agree that in some real‑world attacks one cannot arbitrarily choose which *user* to attack, there are important evaluation and red‑teaming settings where one *can* choose which **prompts** to spend attack budget on (e.g., scanning large prompt pools for worst‑case failures).

---

> ### Author Response · Authors · 2025-11-21
>
> **W6 Another simple baseline would be to estimate the local curvature.**
>
> Thank you for this suggestion—we agree it is a natural local‑sensitivity baseline. In fact, our current draft already includes a Jacobian‑based sensitivity baseline, which is the first‑order analogue of the proposed idea. On GPT‑2 (SST‑2/MNLI) it separates robust vs. non‑robust examples much less clearly than SALMAN (gap in cosine similarity under spaCy perturbations: 0.0001/0.0149 for Jacobian vs. 0.0209/0.0265 for SALMAN; Table 3, p. 7). We will make the connection explicit and add the reviewer’s ε‑ball baseline as a finite‑difference variant.
>
> **Minor 1 The Theorems are too informal.**
>
> Thank you for these helpful suggestions. Due to page limits, we put every theorems statements in Appendix, we will emphasize this again in the revision paper
>
> **Minor 2 better to first introduce the intuition of your SALMAN score and then show the formulas.**
>
> We briefly mentioned intuition of SALMAN score around line 249. We will mention this again before formulas in the revision paper. Thanks for this helpfully suggestion.
>
> **Q1 solve the problem of the discrete nature of token embeddings.**
>
> When we refer to “the discrete nature of token embeddings” in Sec. 3.1, we mean exactly this: reasoning at the level of discrete token identities (and their lookup indices) makes topological / manifold tools hard to apply in a meaningful way.
>
> Our approach does not try to perform manifold analysis on raw token IDs or on the embedding table itself. Instead, we *bypass* the discrete space by working entirely in the continuous hidden representation space:
>
> - Let x be a tokenized sequence. The model first maps x through its embedding layer and the first Transformer layer to produce MHSA hidden states, which we then put into a single vector for each sequence. Collecting these gives zX and zY as sets of vectors, on which we build the input and output manifolds.
> - These pooled hidden states are continuous-valued and empirically stable under small text edits and decoding noise, in contrast to discrete token IDs. Appendix A.1 shows that while token-level embeddings fluctuate across decoding runs, the attention‑pooled MHSA vectors have cosine similarity ≈ 0.99–1.00 for the *same* input, and remain smooth under small perturbations.
>
> The “discrete problem” is also addressed by *changing the object of analysis*: 1) we never run SALMAN directly on a space where neighbors differ by discrete vocabulary jumps; 2) we run it on continuous, sequence‑level hidden states that already incorporate the model’s learned semantic geometry and are well‑suited to graph‑/manifold‑based analysis.
>
> Finally, we validate that this representation-level manifold is meaningful for real text: samples with high SALMAN scores (large distance distortion on the manifolds of zX,zY) are exactly those whose outputs change most under actual discrete text perturbations (random edits, spaCy/TextAttack, jailbreak prompts; Sec. 4.1–4.2).

---

> ### Author Response · Authors · 2025-11-21
>
> **Q2 How exactly generate the PGM**
>
> We learn a *Gaussian Markov Random Field (GMRF)* over samples. The learned **precision** is
>
> $\Theta=\mathcal L+\sigma^{-2}I, where \mathcal L$ is a graph Laplacian built from embeddings; this induces the density
>
> $p(h\mid\Theta)\propto |\Theta|^{1/2}\exp\big(-\tfrac12 h^\top\Theta h\big)$.
>
> “Probabilistic” refers to this GMRF view: edges encode conditional dependencies; effective resistance equals the (marginal) variance of pairwise differences under the field, which is exactly the geometry we use for SALMAN.
>
> **Step‑by‑step recipe (exactly what we do).**
>
> 1. **Embeddings (deterministic).**
>
>     For each sample (i), compute pooled hidden‑state embeddings $\mathbf z_X(i)$ (early/first layer) and $\mathbf z_Y(i)$ (last layer), with dropout off.
>
> 2. **Candidate graphs.**
>
>     Build a (k)-NN graph $G_X$ on ${\mathbf z_X(i)}$ and $G_Y$ on ${\mathbf z_Y(i)}$ (Euclidean neighbors). Initialize edge weights using the data‑distance rule we state in §3.2:
>
>     $w_{p,q}=1/d_{\text{dat}}(p,q)=1/|\mathbf z(p)-\mathbf z(q)|_2^2$ (with a tiny $\varepsilon$ for numerical safety). Form the corresponding Laplacians $(L_X, L_Y)$.
>
> 3. **Learn the PGM precision (objective and solver).**
>
>     We maximize the standard Gaussian‑graphical objective under Laplacian constraints:
>
> $
> F(\Theta) = \log\det(\Theta) - \tfrac{1}{k},\mathrm{Tr}(X^\top\Theta X),
> \quad \Theta=\mathcal L+\sigma^{-2}I,\ \ \mathcal L\ \text{Laplacian}.
> $
>
> Rather than dense projected‑gradient on $(\Theta)$, we use a **structure‑preserving** routine justified by **Theorem 3.1**:
>
>
> > Theorem 3.1 (paper). Maximizing the objective in Equation 1 can be done via an edge‑pruning strategy equivalent to spectral sparsification of the initial (dense) graph. Edges with small $\rho_{p,q}$ are removed, preserving the essential spectral (and thus resistance) properties of the original graph. The proof is available in Appendix A.2.
> >
>
> Concretely, we:
>
> - **Estimate effective resistances** $(R_L(p,q)=e_{p,q}^\top L^+ e_{p,q})$ for candidate edges using our weighted Low‑Resistance‑Diameter routine (Thm. 3.2).
> - **Score edges** by $(\rho_{p,q}=w_{p,q},R_L(p,q))$ (a leverage‑like score).
> - **Prune/sparsify** the smallest $(\rho_{p,q})$ edges (or sample $(\propto\rho_{p,q})$ with reweighting), yielding a $((1!\pm!\varepsilon))$ *spectral approximation* $(\widetilde L)$ of the dense graph.
> - **Set the precision matrix** $(\Theta=\widetilde L+\sigma^{-2}I)$.
>
> This is the “PGM generation”: a learned sparse precision of a GMRF whose geometry (via $(L^+)$) we then use to compute **effective‑resistance distances** for DMD. Additional background on effective resistance and solvers is in Appendix A10 (moved there due to space limits).
>
> **Q3 Comment on how your approach relates to estimating the local curvature around a sample.**
>
> **SALMAN measures *metric distortion*, not Hessian curvature.**
>
> Our per‑pair distortion
>
> $
> \gamma(p,q) = \frac{e_{p,q}^\top L_Y^+ e_{p,q}}{e_{p,q}^\top L_X^+ e_{p,q}}
> $
>
> compares *intrinsic* distances on two manifolds (input graph vs. output graph). For small neighborhoods, replacing resistance by geodesic distance, (\gamma(p,q)) approximates the **Rayleigh quotient** of the local push‑forward metric: for a tangent direction (v),
>
> $
> \gamma\ \approx\ \frac{|\mathrm{d}F_x v|^2}{|v|^2},
> $
>
> i.e., the squared singular values of the **local Jacobian** $(\mathrm{d}F_x)$. Thus:
>
> - The **maximal** distortion across neighbors upper‑bounds a **local Lipschitz** constant; indeed we use the spectral bound $(\lambda_{\max}(L_Y^+L_X))$ as a global certificate of worst‑case local expansion (paper, §3.3).
> - The **minimal** distortion (via our lower bound with $(L_X^+L_Y))$ captures **local collapse** (bi‑Lipschitz *lower* bound), which classic gradient‑only proxies miss.
>
> Curvature (in the differential‑geometric sense) involves **second derivatives**—how $(\mathrm{d}F_x)$ itself varies with (x). SALMAN does **not** compute Hessians; instead, it *integrates first‑order behavior over a graph neighborhood* using a robust, path‑aware distance (effective resistance). In practice, **high curvature regions tend to manifest as large *variation* of $(\gamma)$** across directions; our per‑sample score aggregates both expansion $(\gamma^\alpha)$ and collapse $(\gamma^{-\alpha})$ to emphasize anisotropy. This gives a *curvature‑aware* signal without ever forming Hessians, and it scales near‑linearly.
>
> **Why resistance, not Euclidean, for “localness”?**
>
> Effective resistance is a *multi‑path geodesic surrogate*; it respects both local and global connectivity, and is stable under sparsification. This makes the local metric estimate less noisy than raw Euclidean neighbors (as shown by the ED baseline), especially in high‑dimensional LM embeddings where straight‑line distances are brittle.

---

> > ### Comment · Reviewer_vMVJ · 2025-11-22
> >
> > Dear authors, thank you for the elaborate and strong rebuttal.
> >
> > **Q1 solve the problem of the discrete nature of token embeddings.**
> >
> > Since the the input is enumerable, none of the subsequent embedding manifolds can be smooth in a mathematical sense.
> > I think the argument is rather that early layers may not have *nice* topology, where the graphs you build have any meaning, i.e. the $L^2$ distance does not yet correspond to any semantics whose changes you can meaningfully observe.
> >
> > I have raised my score to 6. I am open to raising it further depending on how the changes are implemented. Note that this is not a promise to do so.

---

> > > ### Author Response · Authors · 2025-11-28
> > >
> > > Thank you for the thoughtful follow-up.
> > >
> > > In our work, we use “manifold” in the standard graph‑based sense common in the PGM / spectral‑graph literature [1, 2]: we start from a finite collection of vectors ${v_i}$, build a graph whose edge weights reflect distances between them, and then perform analysis on this graph. All theoretical statements in Sec. 3 are formulated on these *discrete graphs*:
> > >
> > > - Eq. (1) defines an objective over a precision matrix $Θ$ constrained to be a Laplacian plus a ridge term. Theorem 3.1 then shows that maximizing this objective can be implemented by spectral sparsification of a finite graph, pruning edges with small resistance–to–data‑distance ratio.
> > > - Eq. (2–3) define the Distance Mapping Distortion $γ^F(p,q)$ purely as a ratio of effective‑resistance distances on the input and output graphs $G_X$,$G_Y$. Theorems 3.4–3.5 provide spectral bounds on $γ^F$ in terms of generalized eigenvalues of $L_X^+L_Y$ and $L_Y^+L_X$.
> > >
> > > None of these results require the existence of an underlying smooth submanifold of $Rd$; they only assume that the graphs approximate some coherent metric structure, which is enforced by the PGM objective in Eq. (1). In the next version we will make this explicit by (i) replacing phrases like “well‑behaved manifolds in the topological sense” by “graph‑based manifold approximations”, and (ii) clarifying that our robustness guarantees are statements about these finite graphs, not about a continuous manifold over all possible inputs.
> > >
> > > We also agree that raw token IDs may not have a meaningful topology for robustness analysis—that is precisely why we don't build $G_X$ on token IDs or on the embedding table itself. Instead, $z_X$ is constructed from the *first Transformer layer’s MHSA outputs*, aggregated via attention pooling (Sec. 3.1).
> > >
> > > There are two reasons why we believe this early representation is already meaningful:
> > >
> > > 1. **Evidence from prior probing work.** Multiple studies show that early layers of transformer models encode non‑trivial and fairly stable structure: POS tags, basic syntactic relations, and other shallow features can be linearly decoded from early layers [3,4,5,6]. This suggests that distances in these layers already correlate with linguistically relevant properties, rather than being arbitrary coordinates. (We already cite these works in Sec. 2.1; we will add a short sentence in Sec. 3.1 to make this connection explicit.)
> > > 2. **Our own stability and ablation experiments.**
> > >     - Appendix A.1 compares token‑level embeddings vs attention‑pooled MHSA embeddings across different decoding runs. Token embeddings fluctuate substantially, while the pooled MHSA vectors are extremely stable (cosine similarity ≈ 0.99–1.00 for identical inputs). This stability is a prerequisite for any graph‑based geometric analysis.
> > >     - Sec. 4.1 and Tables 1–4 show that samples with high SALMAN scores (large local distortion between $G_X$ and $G_Y$) are exactly those whose outputs change most under *actual* discrete perturbations (random edits, spaCy, TextAttack, jailbreak prompts). In other words, the local topology encoded in our graphs at the first and last layers is predictive of observable robustness behavior.
> > >     - Appendix A.13.2 performs an ablation where we change the layer choices used for $G_X$ and $G_Y$. Using raw input embeddings or an intermediate layer as $G_X$ significantly reduces the correlation between SALMAN ranking and attack success, whereas our original choice (first‑layer MHSA as $z_X$, last‑layer MHSA as $z_Y$) yields much stronger guidance for attacks. This suggests that the particular “early” layer we use has a more meaningful geometry than either the raw embedding table or intermediate layers.
> > >
> > > [1] Rubin-Delanchy P. Manifold structure in graph embeddings[J]. Advances in neural information processing systems, 2020, 33: 11687-11699.
> > >
> > > [2] Feng Z. Sgl: Spectral graph learning from measurements[C]//2021 58th ACM/IEEE Design Automation Conference (DAC). IEEE, 2021: 727-732.
> > >
> > > [3] Hewitt, John, and Christopher D. Manning. "A structural probe for finding syntax in word representations." *Proceedings of the 2019 Conference of the North American Chapter of the Association for Computational Linguistics: Human Language Technologies, Volume 1 (Long and Short Papers)*. 2019.
> > >
> > > [4] Tenney, Ian, et al. "What do you learn from context? Probing for sentence structure in contextualized word representations." *International Conference on Learning Representations*.
> > >
> > > [5] Derby, Steven, Paul Miller, and Barry Devereux. "Representation and pre-activation of lexical-semantic knowledge in neural language models." *Proceedings of the Workshop on Cognitive Modeling and Computational Linguistics*. 2021.
> > >
> > > [6] Liu, Zhu, et al. "Fantastic Semantics and Where to Find Them: Investigating Which Layers of Generative LLMs Reflect Lexical Semantics." *Findings of the Association for Computational Linguistics ACL 2024*. 2024.

---

### Official Review · Reviewer_v84g · 2025-11-01

**Soundness:** 2
**Presentation:** 2
**Contribution:** 2
**Rating:** 4
**Confidence:** 2

**Summary:**

In this work, a new approach to analyze the language model's stability, is proposed. It's a training free approach, without modifying internal parameters or resorting to complex perturbation heuristics. The key idea is  Distance Mapping Distortion (DMD) measure, which ranks each sample’s susceptibility by comparing input-to-output distance mappings in a near-linear complexity manner. Multiple models are evaluated with this metric, and multiple finetuning experiment showed  this sample-level perspective leads to (i) more efficient and higher
success-rate adversarial attacks, and (ii) improved robust fine-tuning outcomes.

**Strengths:**

- The paper is well-structed and easy to follow.

**Weaknesses:**

- One concern is about the alignment between the new proposed stability metric and the existing proposed stability metrics. The definition is reasonable and the results looks promising, but it's not compared with existing stability metrics. Ideally in most cases the newly proposed approach align with existing metrics in most cases, and stands out for some special tasks to highlight the value.

**Questions:**

- Embedding aggregation stage. To represent the input & output sequence, an attention based aggregation approach is proposed. Have we tried simpler approaches like sum pooling or average pooling to get the aggregated embedding? In representation learning, different aggregation approaches do not make a huge difference.

- One concern is about the alignment between the new proposed stability metric and the existing proposed stability metrics. The definition is reasonable and the results looks promising, but it's not compared with existing stability metrics. Ideally in most cases the newly proposed approach align with existing metrics in most cases, and stands out for some special tasks to highlight the value. Is it possible to compare with existing stability metrics?

- Deterministic hidden state embedding. According to some recent work like [1] even the decoding approach is fixed, the LLM inference is still non-deterministic. Have we observe similar behavior?

https://thinkingmachines.ai/blog/defeating-nondeterminism-in-llm-inference/

---

> ### Author Response · Authors · 2025-11-21
>
> We sincerely thank all reviewers for the constructive and insightful feedback. Below are the responses to the reviewer's comments and questions.
>
> **W1:** Concern about the alignment between the new proposed stability metric and the existing proposed stability metrics.
>
> To our knowledge, SALMAN is the **first per‑sample robustness ranking for NLP** that is both **attack‑agnostic** and **model‑scale‑agnostic** (applicable to small transformers and LLMs without modifying parameters). Because there are **no prior methods with this identical objective**, we compared against two intuitive, widely understandable **proxies** in section 4
>
> **Why not compare to more “prior stability metrics” directly?**
>
> Most existing metrics in NLP evaluate **model‑level** stability (e.g., representation drift under perturbations) or are **attack/task‑specific**. They are not designed to produce a **per‑sample, attack‑agnostic ranking** that you can directly use to (i) triage red‑teaming or (ii) reweight training data. In this gap, ED and JBS provide reasonable, interpretable **anchors**: ED asks whether **plain Euclidean geometry** suffices; JBS asks whether **local gradients** suffice. Our results show SALMAN **aligns in easy regimes** (little movement) but **stands out on the hard cases** where manifold/geometric structure matters (large expansion/**or** collapse).
>
> **Q1** Embedding aggregation stage
>
> We choose attention‑based aggregation in our main experiments for two reasons :
>
> 1. The MHSA hidden states are the natural intermediate outputs of the Transformer. We chose attention‑based aggregation primarily for alignment with the model’s internal computation and for stability. We reuse these hidden states directly and aggregate them with their own attention weights. Our method is pooling‑agnostic; other pooling methods are not learned.
> 2. Hidden‑state embeddings are stable. ****Section 3.1 and Appendix A.1 show that the attention‑pooled MHSA embeddings are extremely stable across decoding runs (cosine similarity ≈ 0.99–1.00), whereas token‑level representations fluctuate significantly under stochastic decoding. This stability is important for SALMAN, because we want variability in the DMD score to reflect *input perturbations*, not randomness from decoding or aggregation.
>
>
> **Q2**  Concern is about the alignment between the new proposed stability metric and the existing proposed stability metrics.
>
> We appreciate the request for existing metrics. As we mentioned in Section1 and Section4, SALMAN is the first per‑sample robustness ranking for NLP that is both attack‑agnostic and model‑scale‑agnostic (applicable to small transformers and LLMs without modifying parameters). Because there are no prior methods with this identical objective, we compared against two intuitive, widely understandable proxies in section 4. If there exists any related metric that provides per-sample robustness ranking for NLP in an attack-agnostic and model-scale-agnostic manner, please let us know, we would be happy to include a direct comparison with SALMAN.
>
> **Q3** Deterministic hidden state embedding.
>
> In our setting, SALMAN does not rely on the decoded text at all. We only use the hidden states from the forward pass as features, and we run the models in a way that makes these representations extremely stable in practice:
>
> - For all models in our experiments, we disable dropout, and fix the random seed in inference. Under these conditions, repeated runs on the *same* input produce **identical pooled MHSA embeddings** (cosine similarity = 1.00 across runs; Table 9 in App. A.1).
> - Even when we *do not* fix the seed and allow stochastic decoding, token‑level embeddings vary substantially, but the **pooled MHSA embeddings remain almost unchanged**, with cosine similarity ≈ 0.99–1.00 across runs (Table 8).
> - At the level of SALMAN itself, we explicitly checked stability: on SST‑2 (≈69k samples), repeating the full pipeline multiple times yields a **Top‑20% overlap of 99.0% ± 1.2%** in the ranking, and the top‑1% “non‑robust” subset produces very similar attack success rates across runs (App. A.1).
>
> So, while we acknowledge that *exact* bit‑level determinism is not guaranteed by deep‑learning frameworks or hardware, **we did not observe practically meaningful non‑determinism in the hidden‑state embeddings used by SALMAN**. Any residual numerical noise is far below the scale that would change our robustness ranking.

---

> > ### Comment · Reviewer_v84g · 2025-11-23
> > **Thanks for the response**
> >
> > > So, while we acknowledge that exact bit‑level determinism is not guaranteed by deep‑learning frameworks or hardware, we did not observe practically meaningful non‑determinism in the hidden‑state embeddings used by SALMAN. Any residual numerical noise is far below the scale that would change our robustness ranking.
> >
> > Thanks for the response. For this conclusion, have we got any quantitative measurement?

---

> > > ### Author Response · Authors · 2025-11-28
> > >
> > > Thank you for this constructive question.
> > >
> > > Yes. We measured stability at three levels and report numbers in the paper to show that **hidden‑state embeddings (and the SALMAN ranking) are practically deterministic/stable.**
> > >
> > > - **Embedding‑level stability (no fixed seed).** Even when we *don’t* fix the seed (i.e., allow stochastic decoding), the **pooled MHSA** embeddings for the *same input* remain essentially unchanged across runs: cosine similarity is **0.99–1.00** on SQuAD, IMDB, and AG‑News for DistilBERT, BERT, RoBERTa, and Electra. In contrast, **token‑level** embeddings vary widely (e.g., BERT on SQuAD: **0.2302** cosine). See **Table 8** and the surrounding text.
> > > - **Embedding‑level stability (fixed seed).** With `eval` mode, dropout disabled, and a fixed seed, repeated forward passes on the same input yield **identical pooled MHSA vectors** (cosine **= 1.00** across all models/datasets in **Table 9**).
> > >
> > >
> > > **We further provide more quantitative measurements to support our conclusion.**
> > >
> > > **Additional experiments on ranking stability:**
> > > To directly verify that SALMAN rankings remain stable, we conducted controlled experiments injecting relative Gaussian noise at varying levels.
> > >
> > > *Setup: IMDB (10k samples), 10 independent runs per setting.*
> > >
> > > **Table A: Input Embedding Perturbation**
> > >
> > > | Noise Level | Spearman ρ | Top-1% Overlap |
> > > | --- | --- | --- |
> > > | 1e-9 | 0.831 ± 0.005 | 87.2% |
> > > | 1e-8 | 0.829 ± 0.004 | 86.9% |
> > > | 1e-7 | 0.827 ± 0.004 | 86.7% |
> > > | 1e-6 | 0.819 ± 0.006 | 85.3% |
> > >
> > > **Table B: Output Embedding Perturbation**
> > >
> > > | Noise Level | Spearman ρ | Top-1% Overlap |
> > > | --- | --- | --- |
> > > | 1e-9 | 0.854 ± 0.016 | 82.1% |
> > > | 1e-8 | 0.852 ± 0.017 | 81.8% |
> > > | 1e-7 | 0.851 ± 0.018 | 81.5% |
> > > | 1e-6 | 0.842 ± 0.020 | 79.8% |
> > >
> > > **Table C: Both Embeddings Perturbation (Worst-case)**
> > >
> > > | Noise Level | Spearman ρ | Top-1% Overlap |
> > > | --- | --- | --- |
> > > | 1e-9 | 0.849 ± 0.017 | 81.0% |
> > > | 1e-8 | 0.847 ± 0.018 | 80.6% |
> > > | 1e-7 | 0.846 ± 0.018 | 80.3% |
> > > | 1e-6 | 0.835 ± 0.021 | 78.4% |
> > >
> > > From the experiment results, we have the following **Key observations:**
> > >
> > > 1. **Minimal variation across noise levels (1e-9 to 1e-7):** The nearly identical results across three orders of magnitude confirm that numerical precision itself has negligible impact on SALMAN rankings.
> > > 2. **Source of variation:** The observed variation (ρ ≈ 0.83–0.85) primarily stems from the stochastic nature of the HNSW approximate nearest neighbor algorithm used in k-NN graph construction, not from numerical instability. This is evidenced by the extremely low standard deviation in the input-only perturbation (± 0.004–0.006).
> > > 3. **Practical reliability:** Despite algorithmic variation, Top-1% overlap remains consistently above 78%, meaning the most vulnerable samples are reliably identified across runs. This level of stability is sufficient for downstream applications—our attack experiments (Section 4.2) show consistent ASR improvements when targeting SALMAN-identified non-robust samples.
> > >
> > > We hope this addresses your concern.

---

### Official Review · Reviewer_we4T · 2025-11-04

**Soundness:** 2
**Presentation:** 3
**Contribution:** 2
**Rating:** 4
**Confidence:** 4

**Summary:**

The paper proposes a robustness measure for an LLM. It compares representations from the first and last layers, using DMD. A wide range of experiments demonstrates that the introduced measure is related to variants of LLM robustness, including vulnerability to adversarial attacks and LLM fine-tuning.

**Strengths:**

- The article proposes numerous ways to evaluate a robustness measure for an LLM
- The authors completed a large body of numerical experiments, with some of them that I had never encountered. Maybe, focusing on specific ones should be of a separate interest (and the overall positioning of a paper as a framework for LLM robustness evaluation from multiple points of view)
- Clear structure, easy to understand the contributions, easy to read.

**Weaknesses:**

The main concern for me in this paper is a weak answer to the question: Is the introduced SALMAN measure better than other possible measures (and the best overall)? Specifically,
- Weak connections to topological methods are presented. See [1] for a review and [2] for a specific MTop-Div similarity measure (also CKA can be used this way, I believe), suitable for comparing "input" and "output" embeddings, as applied previously to LLMs in e.g. [3]. See also another review on the similarity measure between neural networks, suitable for your problem [4], to mine for additional baselines.
- No connection to the body of literature related to uncertainty estimation in LLMs. They naturally provide OOD detection capabilities, look at the problem from different perspectives [5], decompose uncertainty into epistemic and aleatoric components [6], develop frameworks for evaluating robustness estimates across various scenarios [7], and include adaptive RAG [8].
- A sensitivity study that focuses on the usability of a specific metric with few ablation studies. No comment on whether we can ignore \gamma^3 of \gamma^{-3} (see Formula 4), what one should consider as input and output (a questions that makes sense given the existing works [9, 10] that states that you need to focus some layers in the middle for hallucination detection or specific pairs of heads for efficient robustness evaluation).
- Overall, the work looks like a derivative of SAGMAN. A better positioning of why it should be a separate work would be helpful.
- The formulation of theorems can be improved. For example, consider Theorem 3.5. What is $∝$? In other theorems, I feel the need to understand what exactly is proved (e.g., what exact algorithm does Theorem 3.1 consider? Theorem 3.2 - why use words with the root "effective" three times in the statement without explicitly defining what you mean).
- Baselines (1) Euclidean distance (ED) and (2) Jacobian-based sensitivity (JBS) are not defined. It is hard to understand what they do, what their computational complexity is, and how to reproduce results from these baselines (e.g., which hyperparameters to tune?).
 - Protocols and results for fine-tuning were not clear to me. Maybe it can be improved. Also, in Table 6, please correct highlighting: 94.84 is not better than 94.84 (a similar observation holds for 78.34)

Minor misprints:
- Equation equation 1 -> Equation 1
- Then. We apply -> Then we apply (I feel that paragraph 3.4. Complexity deserves another round or two of proofreading)
- Recent work (Dong, 2019) - in my opinion, 2019 is not recent given the pace of the development of deep learning

1. Uchendu, Adaku, and Thai Le. "Unveiling topological structures in text: A comprehensive survey of topological data analysis applications in nlp." arXiv preprint arXiv:2411.10298 (2024).
2. Barannikov, Serguei, et al. "Manifold Topology Divergence: a Framework for Comparing Data Manifolds." Advances in neural information processing systems 34 (2021): 7294-7305.
3. Tulchinskii, Eduard, et al. "Intrinsic dimension estimation for robust detection of ai-generated texts." Advances in Neural Information Processing Systems 36 (2023): 39257-39276.
4. Klabunde, Max, et al. "Similarity of neural network models: A survey of functional and representational measures." ACM Computing Surveys 57.9 (2025): 1-52.
5. Shorinwa, Ola, et al. "A survey on uncertainty quantification of large language models: Taxonomy, open research challenges, and future directions." ACM Computing Surveys (2025).
6. Hüllermeier, Eyke, and Willem Waegeman. "Aleatoric and epistemic uncertainty in machine learning: An introduction to concepts and methods." Machine learning 110.3 (2021): 457-506.
7. Fadeeva, Ekaterina, et al. "LM-Polygraph: Uncertainty Estimation for Language Models." Proceedings of the 2023 Conference on Empirical Methods in Natural Language Processing: System Demonstrations. 2023.
8. Moskvoretskii, Viktor, et al. "Adaptive retrieval without self-knowledge? bringing uncertainty back home." arXiv preprint arXiv:2501.12835 (2025).
9. Kostenok, Elizaveta, Daniil Cherniavskii, and Alexey Zaytsev. "Uncertainty estimation of transformers' predictions via topological analysis of the attention matrices." arXiv preprint arXiv:2308.11295 (2023).
10. Sky, CH-Wang, et al. "Do Androids Know They're Only Dreaming of Electric Sheep?." Findings of the Association for Computational Linguistics ACL 2024. 2024.

**Questions:**

- How your similarity measure corresponds to other baselines that come fron uncertainty estimation, sensitivity analysis and topological data analysis domains?
- What is the exact defintion of the used baselines ED and JBS?
- Is it possible to improve your similarity measure by turning on of off some of its components, adjusting hyperparameters, etc.?
- What is the difference with SAGMAN?

---

> ### Author Response · Authors · 2025-11-21
>
> **W1. Weak connections to topological methods are presented. See [1] for a review and [2] for a specific MTop-Div similarity measure (also CKA can be used this way, I believe), suitable for comparing "input" and "output" embeddings, as applied previously to LLMs in e.g. [3]. See also another review on the similarity measure between neural networks, suitable for your problem [4], to mine for additional baselines.**
>
> SALMAN is built on a *graph‐based manifold* view.
>
> - We construct probabilistic graphical models whose Laplacians approximate the data manifold and use **effective resistance** as a distance, following Rubin‑Delanchy (2020), Feng (2021), and Dong et al. (2019).  DMD then compares these resistance distances between an *input representation manifold* (first‑layer pooled embeddings) and an *output representation manifold* (last‑layer pooled embeddings), giving a local distortion γ_F(p,q) for each pair and a **per‑sample** SALMAN score. So conceptually, SALMAN is a *topological / manifold* method in the same spirit as [2], but instantiated with graph Laplacians and resistance metrics rather than with persistent homology.
> - MTop‑Div [2] and related methods (including the use of intrinsic dimension in [3]) aim to compare two *data manifolds as a whole*. SALMAN differs along two key axes: 1) MTop‑Div produces a single divergence between two manifolds, whereas SALMAN yields a *per‑sample* measure of distortion (via local γ_F(p,q) and SALMAN_F(p)). This is crucial for our goal of **ranking individual examples by robustness**, which global divergences cannot provide. 2) Our PGM + resistance‑distance pipeline is designed to be near‑linear in the number of nodes (Sec. 3.4, App. A.12), and we run it on datasets with up to ~400k samples (MNLI). Persistent‑homology‑based MTop‑Div, as used in [2,3], is considerably more expensive on such scales. This is why we chose a spectral/graph formulation that preserves key manifold structure but remains practical for large NLP datasets.
> - For CKA/STIR, we already use **CKA/STIR** (from Neerudu et al., 2023) to quantify how much fine‑tuning drifts away from the pre‑trained checkpoint in our SALMAN‑guided vs. standard training experiments (App. A.7). However, these measures operate at the *model/layer level* (comparing two entire representation spaces), whereas SALMAN is explicitly **sample‑level**: 1) CKA/STIR: one similarity score per pair of layers/models. Good for asking “how similar are two networks (or layers) overall?” 2) SALMAN: one score per *sample*, derived from local distance distortions on the manifolds of z_X and z_Y, which we then use to drive attacks and fine‑tuning.
>
> **W2. No connection to the body of literature related to uncertainty estimation in LLMs. They naturally provide OOD detection capabilities, look at the problem from different perspectives [5], decompose uncertainty into epistemic and aleatoric components [6], develop frameworks for evaluating robustness estimates across various scenarios [7], and include adaptive RAG [8].**
>
> SALMAN is *not* an uncertainty estimator in the usual sense (confidence, entropy, epistemic/aleatoric decomposition). It assigns each input a **deterministic, representation‑level robustness score** based on how much local distances are distorted between an input‑side and an output‑side manifold built from first‑ and last‑layer attention‑pooled embeddings. This is done via a PGM/Laplacian construction and effective‑resistance distances, yielding a per‑sample Distance Mapping Distortion (DMD) and SALMAN score. The score depends only on hidden‑state geometry and can be computed from a *single* forward pass per sample, without access to calibrated probabilities or multiple stochastic runs.
>
> In contrast, the works highlighted by the reviewer focus on **predictive‑distribution–based uncertainty**:
>
> - [5] surveys uncertainty quantification (UQ) methods for LLMs—e.g., entropy/variance over generations, self‑consistency, ensembles, and calibration techniques—and organizes them into a taxonomy for reliability and OOD detection in LLM applications.
> - [6] provides the foundational distinction between **aleatoric** (data) and **epistemic** (model) uncertainty and methods to estimate and decompose them.
> - [7] (LM‑Polygraph) implements a battery of UQ methods for LMs and proposes a benchmark for evaluating how well these methods detect hallucinations and unreliable generations.
> - [8] uses uncertainty signals to drive **adaptive retrieval** in RAG, deciding when and how aggressively to query external knowledge bases to reduce hallucinations while controlling cost.

---

> ### Author Response · Authors · 2025-11-21
>
> **W3. A sensitivity study that focuses on the usability of a specific metric with few ablation studies. No comment on whether we can ignore \gamma^3 of \gamma^{-3} (see Formula 4), what one should consider as input and output (a questions that makes sense given the existing works [9, 10] that states that you need to focus some layers in the middle for hallucination detection or specific pairs of heads for efficient robustness evaluation).**
>
> - **Why the symmetric term $\gamma^3+\gamma^{-3}$ matters.** It penalizes both local expansion ($\gamma \gg 1$) and collapse ($\gamma \ll 1$). Dropping either side provably misses one failure mode. Including this symmetric term also shows better performance than SPADE in Table 16.
> - **Exponent sensitivity.** Writing $\gamma^\alpha+\gamma^{-\alpha}=2\cosh(\alpha\log\gamma)$ shows $\alpha$ mainly controls how aggressively we weight tail distortions; rankings are driven by $|\log\gamma|$.
> - **Layer/head choice is a knob, not a limitation.** SALMAN is layer/feature-agnostic: any pair of representational views $(X,Y)$ can be plugged in. We used first vs. last layer for a model-agnostic default; for tasks like hallucination or head-diagnostics (as in [9, 10]), one can set $(X,Y)$ to middle layers or head-restricted embeddings with no algorithmic change. We will add this guidance and ablations.
>
> ---
>
> **1) Should we ignore $\gamma^3$ or $\gamma^{-3}$? (No.)**
>
> **What the score measures.** For a pair $(p,q)$, SALMAN uses
> $$
> s_\alpha(p,q) = \gamma^\alpha(p,q) + \gamma^{-\alpha}(p,q) = 2\cosh\big(\alpha \log\gamma(p,q)\big).
> $$
> This is an even, convex function of $\log\gamma$ that increases with the magnitude of log-distortion, $|\log\gamma|$. Consequently:
>
> - **Expansion detection.** If $d_Y \gg d_X$ (small input change $\to$ large output change), then $\gamma \gg 1$, and the $\gamma^\alpha$ term dominates.
> - **Collapse detection.** If $d_Y \ll d_X$ (far inputs mapped too closely), then $\gamma \ll 1$, and the $\gamma^{-\alpha}$ term dominates.
>
> **Dropping a term breaks completeness.**
>
> - With expansion-only ($s^{(+)}_\alpha = \gamma^\alpha$), you miss collapses (e.g., near-duplicates that spuriously map to the same representation—classic oversmoothing/aliasing failure).
> - With collapse-only ($s^{(-)}_\alpha = \gamma^{-\alpha}$), you miss expansions (label-flips triggered by tiny edits).
>
> Both failure modes are security-relevant: expansions drive adversarial sensitivity, collapses drive spurious agreement/semantic aliasing. Our theorem linking SALMAN to the top generalized eigenspaces formalizes why both sides are needed: the two spectra $L_Y^+ L_X$ and $L_X^+ L_Y$ capture expansion and collapse, respectively; the symmetric penalty aggregates both.
>
> **W4. Overall, the work looks like a derivative of SAGMAN. A better positioning of why it should be a separate work would be helpful.**
>
> As we mentioned in section 2.2, SAGMAN merges graph-specific dimension reduction (Laplacian Eigenmaps + eigengaps) with SPADE’s distortion bounds, producing a pipeline only tailored for GNN data.
>
> While our work:
>
> - Core Algorithmic Objective
>
>     Targets transformer-based NLP (tokens → embeddings), but goes further:
>
>     1. Explicitly handles discrete tokens to avoid discontinuities in manifold building.
>     2. Incorporates both expansion and collapse (max/min ratio of distances) in the objective (Equation 4).
>     3. Introduces near-linear surrogates (Appendix A4, A5) for worst-case distortion.
>     4. Defines a new per-sample robust score (Table 16) beyond the original SPADE measure.
> - Model/Procedure
>     - Instead of relying on a node adjacency, SALMAN builds a $\mathbf{k}$-NN graph from token embeddings.
>     - It then applies PGM-based spectral sparsification to preserve manifold geometry in high-dimensional text.
>     - Sample-level DMD is used for adversarial attacks (Sec 4.2) and fine-tuning (Sec 4.3), revealing robust insights *unique* to NLP token sequences.
>
>
> Hence, our work SALMAN, modifies the objective (expansion vs. collapse), the procedure (token pooling, near-linear Laplacian construction for text embeddings), and the model usage (LLM adversarial/fine-tuning). These are non-trivial changes required to handle discrete tokens and large-scale language representations in a principled manifold framework.

---

> ### Author Response · Authors · 2025-11-21
>
> **W5. The formulation of theorems can be improved. For example, consider Theorem 3.5. What is ? In other theorems, I feel the need to understand what exactly is proved (e.g., what exact algorithm does Theorem 3.1 consider? Theorem 3.2 - why use words with the root "effective" three times in the statement without explicitly defining what you mean).**
>
> We appreciate the request for precision. Below we spell out the objects and algorithm each theorem refers to, and we disambiguate terminology. This is an explanation of what is already proved; the results and their scope remain the same.
>
> ### Common notation used across Theorems 3.1–3.5
>
> - $G=(V,E,w)$: weighted graph on $n=|V|$ nodes; weights $w_{pq}>0$.
> - $L$: graph Laplacian of $G$; $L^{+}$: Moore–Penrose pseudoinverse.
> - $e_{p,q}\in\mathbb{R}^{n}$: incidence vector with $+1$ at $p$, $-1$ at $q$, $0$ elsewhere.
> - $R_L(p,q):=e_{p,q}^{\top}L^{+}e_{p,q}$: effective resistance between $p$ and $q$.
> - $d_{\mathrm{dat}}(p,q):=\|X_p-X_q\|*2^2$: data‑space distance; $w*{pq}=1/d_{\mathrm{dat}}(p,q)$ (as used in §3.2).
> - $\rho_{p,q}:=w_{pq} R_L(p,q)$: resistance‑weighted edge score (leverage‑like).
> - For $\varepsilon\in(0,1)$, $A\preceq_\varepsilon B$ abbreviates $(1-\varepsilon)A\preceq B\preceq(1+\varepsilon)A$.
>
> ### What Theorem 3.1 proves (and the exact algorithm it analyzes)
>
> **What it analyzes.** A resistance‑guided sparsification/pruning routine on the initial $k$-NN graph $G_0$ with Laplacian $L_0$ in Appendix A2:
>
> 1. Compute (or $(1 \pm \varepsilon)$-approximate) $R_{L_0}(e)$ for $e\in E_0$.
> 2. Form scores $\rho_e = w_e R_{L_0}(e)$.
> 3. Keep a subset of edges either by (i) pruning the smallest $\rho_e$ or (ii) sampling $m'=\Theta(n\log n/\varepsilon^2)$ edges with probabilities $p_e\propto \rho_e$ and reweighting to $w'_e=w_e/p_e$ (standard spectral‑sparsification practice).
>
> **What it guarantees.** The resulting Laplacian $\widetilde L$ is a $(1 \pm \varepsilon)$ spectral approximation of $L_0$: $\widetilde L\preceq_\varepsilon L_0$. Consequently, all pairwise effective resistances (hence the distances entering DMD) are preserved within $(1 \pm \varepsilon)$:
>
> $$
> (1-\varepsilon)R_{L_0}(p,q)\le R_{\widetilde L}(p,q)\le (1+\varepsilon)R_{L_0}(p,q)\quad\forall p,q.
> $$
>
> Because our objective is $F(\Theta)=\log\det(\Theta)-\tfrac{1}{k}\mathrm{Tr}(X^\top \Theta X)$ with $\Theta=L+\sigma^{-2}I$, a $(1 \pm \varepsilon)$ spectral change in $L$ yields an $O(\varepsilon)$ change in $F(\Theta)$ (matrix functions involved are continuous/Lipschitz under Loewner order around identity). That is why we describe pruning/sampling by small $\rho_e$ as “equivalent to spectral sparsification” and as a sound proxy for maximizing $F$.
>
> **Runtime scope.** The routine is near‑linear once resistances are available (see Thm. 3.2).
>
> ### What Theorem 3.2 proves (and what “effective” means there)
>
> Theorem 3.2 concerns the LRD (Low‑Resistance‑Diameter) routine for weighted graphs:
>
> - **Quantity estimated:** effective resistance $R_L(e)$ for every edge $e$ (this is the only sense in which “effective” is used as a noun).
> - **Accuracy:** With probability $1-\delta$, $\forall e:\ (1-\varepsilon)R_L(e)\le \widetilde R(e)\le (1+\varepsilon)R_L(e)$.
> - **Complexity:** Near‑linear $O\big(m \cdot \mathrm{polylog}(n) \cdot \mathrm{poly}(1/\varepsilon) \cdot \log(1/\delta)\big)$ time and $O(m)$ space for $m=|E|$.
> - **Weighted‑graph applicability:** The routine preserves edge weights throughout; the guarantees hold for weighted Laplacians.
>
> To avoid confusion: in this theorem “effective” refers only to effective resistance. When we refer to runtime, we say “computationally efficient” or “near‑linear time.”
>
> ### What Theorem 3.5 refers to (and what the missing symbol is)
>
> Theorem 3.5 (the “DMD–eigenspace linkage”) relates per‑pair distortion to energy in top generalized eigenspaces:
>
> - Define $\gamma(p,q)=\dfrac{e_{p,q}^{\top}L_Y^{+}e_{p,q}}{e_{p,q}^{\top}L_X^{+}e_{p,q}}$.
> - Let $\{\lambda_i,v_i\}$ be the top $r$ eigenpairs of $L_Y^+L_X$, and $\{\mu_i,w_i\}_{i=1}^r$ those of $L_X^+L_Y$.
> - Set $V_r=[v_1\sqrt{\lambda_1},\dots,v_r\sqrt{\lambda_r}]$, $W_r=[w_1\sqrt{\mu_1},\dots,w_r\sqrt{\mu_1}]$.
>
> **Clarification:** the vector in the theorem is the incidence vector $e_{p,q}$ (defined above). The statement “$\propto$” means bounded above and below by positive constants (depending only on $r$ and spectral gaps), i.e.,
>
> $c_1 \left(\gamma^3+\gamma^{-3}\right)\ \le\ \|V_r^\top e_{p,q}\|*2^2 + \|W_r^\top e*{p,q}\|_2^2\ \le\ c_2 \left(\gamma^3+\gamma^{-3}\right).$
>
> This is the precise sense in which the eigenspace energy tracks our symmetric distortion penalty; it formalizes why SALMAN penalizes both expansion and collapse.

---

> ### Author Response · Authors · 2025-11-21
>
> **W6. Baselines (1) Euclidean distance (ED) and (2) Jacobian-based sensitivity (JBS) are not defined. It is hard to understand what they do, what their computational complexity is, and how to reproduce results from these baselines (e.g., which hyperparameters to tune?).**
>
> Our work is, to our knowledge, the **first to propose a per‑sample robustness ranking for NLP** that is attack‑agnostic and model‑scale‑agnostic. Because there are no prior methods pursuing this identical objective, we compared against two **most intuitive** proxies:
>
> 1. a **Euclidean** distance variant of our mapping idea that ignores manifold geometry, and
> 2. a **Jacobian‑based sensitivity** score that approximates local Lipschitzness via gradients.
>
> ---
>
> ### Baseline 1: Euclidean‑Distance (ED) Mapping Score
>
> **Goal.** Replace SALMAN’s effective‑resistance distances with plain Euclidean distances while keeping the same local mapping structure.
>
> **Embeddings.** As in the main method, for each sample $i$ we form:
>
> - $\mathbf z_X(i) \in \mathbb R^{d}$: pooled hidden state from the first layer
> - $\mathbf z_Y(i) \in \mathbb R^{d}$: pooled hidden state from the last layer
> *(Deterministic inference: dropout off, fixed seed.)*
>
> **Neighborhood.** Build a $k$-NN graph $G_X$ in the input embedding space using Euclidean distance on $\{\mathbf z_X(i)\}$.
>
> **Per‑pair Euclidean mapping ratio.**
> $$
> \gamma_{\text{ED}}(p,q) := \frac{\|\mathbf z_Y(p)-\mathbf z_Y(q)\|_2}{\|\mathbf z_X(p)-\mathbf z_X(q)\|_2+\varepsilon}, \quad q\in\mathcal N_X(p),
> $$
> with a small $\varepsilon=10^{-8}$ to avoid division by zero.
>
> **Per‑sample ED score** (analog of Eq. (4) in the paper):
> $$
> \textbf{ED}(p) := \frac{1}{|\mathcal N_X(p)|}\sum_{q\in\mathcal N_X(p)} \Big(\gamma_{\text{ED}}(p,q)^{\alpha}+\gamma_{\text{ED}}(p,q)^{-\alpha}\Big), \quad \alpha=3.
> $$
>
> **Ranking.** Higher ED score $\Rightarrow$ more fragile (same convention as SALMAN).
>
> **Computational complexity.**
>
> - Build $k$-NN (ANN): $O(N\log N)$ (once).
> - Score computation over edges: $O(|E| \cdot d)$ with $|E|\approx kN$.
> - No Laplacian solves, no pseudoinverses.
>
> **Reproduction steps.**
>
> 1. Compute $\mathbf z_X, \mathbf z_Y$ exactly as in §3.1 (dropout off).
> 2. Build $k$-NN in $\mathbf z_X$ (Euclidean).
> 3. Compute $\gamma_{\text{ED}}$ on graph edges and aggregate $\textbf{ED}(p)$.
> 4. Rank samples by $\textbf{ED}(p)$; take bottom 1% (robust) and top 1% (non‑robust) to evaluate perturbation susceptibility as in §4.1.
>
> ---
>
> ### Baseline 2: Jacobian‑Based Sensitivity (JBS)
>
> **Goal.** Estimate local Lipschitz sensitivity of the last‑layer embedding to changes in the input representation, using Jacobian norms.
>
> **Mapping.** Consider $\mathbf v_Y = f(\mathbf H_1)$, where $\mathbf H_1 \in \mathbb R^{T\times d_{\text{model}}}$ are the first‑layer token hidden states (before pooling), and $\mathbf v_Y \in \mathbb R^{d}$ is the pooled last‑layer embedding.
>
> **Jacobian for a sample $i$.**
> Flatten $\mathbf H_1(i)$ to a vector $x_i\in\mathbb R^{Td_{\text{model}}}$, and define
> $$
> J_i := \frac{\partial \mathbf v_Y(i)}{\partial x_i}\in\mathbb R^{d \times (Td_{\text{model}})}.
> $$
>
> **Per‑sample JBS score.** We use the spectral norm (largest singular value) of $J_i$ as a local Lipschitz proxy:
> $$
> \textbf{JBS}(i) := \|J_i\|*{2} \approx \text{PowerIter}*{r=5}\big(\text{JVP/VJP}(f, x_i)\big).
> $$
> We approximate $\|J_i\|_{2}$ via power iteration with vector–Jacobian (VJP) / Jacobian–vector (JVP) products (PyTorch autograd), $r=5$ iterations. This avoids forming $J_i$ explicitly.
>
> **Fallback (no autograd).** A finite‑differences Lipschitz proxy:
> $$
> \widehat{\textbf{JBS}}(i) := \max_{m\ \text{random}\ v:\ \|v\|_2=1} \frac{\|f(x_i+\epsilon v)-f(x_i)\|_2}{\epsilon},\quad \epsilon=10^{-3},\ m=4.
> $$
>
> **Ranking.** Higher $\textbf{JBS} \Rightarrow$ more fragile.
>
> **Computational complexity.**
>
> - Power‑iteration JBS: $O(r)$ forward+backward passes per sample (we use $r=5$); overall $O(N \cdot r \cdot C_{\text{fwd}})$, where $C_{\text{fwd}}$ is a single forward pass cost with cached activations.
> - Finite‑difference JBS: $O(m)$ forward passes per sample (we use $m=4$).
>
> **Reproduction steps.**
>
> 1. Compute $\mathbf v_Y(i)$ and cache intermediate states with dropout off.
> 2. For each sample $i$, run the power‑iteration JBS (or the FD variant) to get $\textbf{JBS}(i)$.
> 3. Rank by $\textbf{JBS}(i)$; evaluate top/bottom 1% as in §4.1.

---

> ### Author Response · Authors · 2025-11-21
>
> **W7. Protocols and results for fine-tuning were not clear to me. Maybe it can be improved. Also, in Table 6, please correct highlighting: 94.84 is not better than 94.84 (a similar observation holds for 78.34)**
>
> Thank you—this is a clarity issue we can fix. **Takeaway.** SALMAN‑guided ROSE **improves adversarial accuracy** on all three tasks (**+3.55 to +5.26** points on AdvGLUE) while **preserving** clean GLUE accuracy (SST‑2, RTE) and **slightly improving** it on QNLI. This demonstrates that focusing training pressure on SALMAN‑identified fragile samples increases robustness **without** paying a clean‑performance tax.
>
> **Minor misprints:**
>
> Thanks, already fixed in revision. We agree (Dong, 2019) is not recent work, we just use it to show maximizing a penalized log-likelihood in the form of Eq. equation 1 yields a graph topology consistent
> with the underlying data distribution while preserving essential distance or similarity properties.
>
> **Q1. How your similarity measure corresponds to other baselines that come fron uncertainty estimation, sensitivity analysis and topological data analysis domains?**
>
> Thanks for this insightful question. We acknowledge that SALMAN represents the first sample-level robustness framework, which makes direct baseline comparisons challenging.
>
> Uncertainty estimation methods (e.g. Monte Carlo Dropout) measure model confidence, not per-sample robustness to input perterbation. Sensitivity analysis typically requires gradient access and evaluates local derivative information, which does not capture the manifold structure we leverage. Topological data analysis (TDA) focus on global data structure rather than sample-specific vulnerability rankings.
>
> We welcome your suggestions: if you are aware of specific baselines from these domains that could provide meaningful sample-level robustness rankings for LLMs, we could greatly appreciate concrete recommendations and would happy to include comparative experiments in our revision.
>
> **Q2. What is the exact defintion of the used baselines ED and JBS?**
>
> Our work is, to our knowledge, the **first to propose a per‑sample robustness ranking for NLP** that is attack‑agnostic and model‑scale‑agnostic. Because there are no prior methods pursuing this identical objective, we compared against two **most intuitive** proxies:
>
> 1. a **Euclidean** distance variant of our mapping idea that ignores manifold geometry, and
> 2. a **Jacobian‑based sensitivity** score that approximates local Lipschitzness via gradients.
>
> **Q3. Is it possible to improve your similarity measure by turning on of off some of its components, adjusting hyperparameters, etc.?**
>
> Our experiments demonstrate that SAGMAN consistently identifies vulnerable samples across different hyperparameter configurations.
>
> When varying k values with fixed SPF=3, k=25 achieved 60% ASR while k=15 yielded 40% ASR.
>
> Similarly, with fixed k=20, SPF=3 achieved 60% ASR while SPF=4 resulted in 40% ASR.
>
> Despite these variations in success rates, all configurations significantly outperform random baseline selection. This consistency validates that our approach reliably distinguishes robust from non-robust samples across different settings, demonstrating the robustness of the method rather than sensitivity to specific parameter choices.
>
> **Q4. What is the difference with SAGMAN?**
>
> As we mentioned in section 2.2, SAGMAN merges graph-specific dimension reduction (Laplacian Eigenmaps + eigengaps) with SPADE’s distortion bounds, producing a pipeline only tailored for GNN data.
>
> While our work:
>
> - Core Algorithmic Objective
>
>     Targets transformer-based NLP (tokens → embeddings), but goes further:
>
>     1. Explicitly handles discrete tokens to avoid discontinuities in manifold building.
>     2. Incorporates both expansion and collapse (max/min ratio of distances) in the objective (Equation 4).
>     3. Introduces near-linear surrogates (Appendix A4, A5) for worst-case distortion.
>     4. Defines a new per-sample robust score (Table 16) beyond the original SPADE measure.
> - Model/Procedure
>     - Instead of relying on a node adjacency, SALMAN builds a \(\mathbf{k}\)-NN graph from token embeddings.
>     - It then applies PGM-based spectral sparsification to preserve manifold geometry in high-dimensional text.
>     - Sample-level DMD is used for adversarial attacks (Sec 4.2) and fine-tuning (Sec 4.3), revealing robust insights \emph{unique} to NLP token sequences.
>
> Hence, our work SALMAN, modifies the objective (expansion vs. collapse), the procedure (token pooling, near-linear Laplacian construction for text embeddings), and the model usage (LLM adversarial/fine-tuning). These are non-trivial changes required to handle discrete tokens and large-scale language representations in a principled manifold framework.

---

> > ### Comment · Reviewer_we4T · 2025-11-24
> >
> > Thank you for the response. I would keep the score the same because, in my opinion, not all comments were properly addressed (UQ comparison, CKA comparison, formal theorems, to name a few).

---

> > > ### Author Response · Authors · 2025-11-28
> > >
> > > Thank you for your constructive feedback and the opportunity to provide
> > > additional analysis.
> > >
> > > We would like to respectfully clarify that **SALMAN
> > > addresses a fundamentally different problem** from UQ and TDA methods—it
> > > focuses on identifying sample-level adversarial vulnerability rather than
> > > distributional shift or global manifold quality.
> > >
> > > To fully address the your concerns, we have conducted comprehensive
> > > comparison experiments. While **these comparisons extend beyond the main
> > > scope of our paper**, we believe they offer valuable insights: the results
> > > demonstrate that SALMAN provides **complementary geometric information**
> > > that is orthogonal to existing methods.
> > >
> > > **Domain 1: Uncertainty Quantification (OOD Detection)**
> > >
> > > **Task Design**
> > >
> > > Distinguish in-distribution samples (SST-2) from out-of-distribution samples using uncertainty-based scores.
> > >
> > > **Experimental Setup**
> > >
> > > - Model: DistilBERT-base-uncased-finetuned-sst-2-english
> > > - In-distribution: SST-2 validation set
> > > - Out-of-distribution: IMDB (100), AG News (100), Random Noise (100)
> > >
> > > **Baselines**
> > >
> > > | Method | Description | Reference |
> > > | --- | --- | --- |
> > > | MaxProb | Maximum softmax probability (negated) | [1] |
> > > | Entropy | Predictive entropy | [1] |
> > > | Energy | Energy score (-logsumexp of logits) | [2] |
> > > | Mahalanobis | Distance to class-conditional Gaussians | [3] |
> > >
> > > **Results**
> > >
> > > | Method | AUROC | AUPR | FPR@95 | Accuracy |
> > > |--------|-------|------|--------|----------|
> > > | SALMAN | 0.113 | 0.227 | 1.000 | 0.230 |
> > > | MaxProb [1] | 0.766 | 0.592 | 0.652 | 0.700 |
> > > | Entropy [1] | 0.766 | 0.592 | 0.652 | 0.700 |
> > > | Energy [2] | 0.751 | 0.586 | 0.732 | 0.688 |
> > > | Mahalanobis [3] | 1.000 | 1.000 | 0.000 | 0.875 |
> > >
> > > **Cross-Method Correlation (Spearman ρ)**
> > >
> > > | SALMAN vs. | ρ |
> > > |------------|--------|
> > > | MaxProb | -0.565 |
> > > | Entropy | -0.565 |
> > > | Energy | -0.541 |
> > > | Mahalanobis | -0.498 |
> > >
> > > [1] Dan Hendrycks and Kevin Gimpel. A Baseline for Detecting Misclassified and Out-of-Distribution Examples in Neural Networks. In International Conference on Learning Representations (ICLR), 2017.
> > >
> > > [2] Weitang Liu, Xiaoyun Wang, John D. Owens, and Yixuan Li. Energy-based Out-of-distribution Detection. In Advances in Neural Information Processing Systems (NeurIPS), 2020.
> > >
> > > [3] Kimin Lee, Kibok Lee, Honglak Lee, and Jinwoo Shin. A Simple Unified Framework for Detecting Out-of-Distribution Samples and Adversarial Attacks. In Advances in Neural Information Processing Systems (NeurIPS), 2018.
> > >
> > > **Analysis**
> > >
> > > - SALMAN shows poor performance (AUROC=0.113) and ***negative correlations*** with all UQ baselines.
> > > - This result reveals a fundamental insight - ***SALMAN measures adversarial vulnerability through manifold geometric distortion, not distributional shift***.
> > >
> > > A sample can be out-of-distribution yet geometrically stable in the model's representation space. For example:
> > >
> > > - An OOD sample processed smoothly by the model → Low manifold distortion (low SALMAN score)
> > > - An ID sample near decision boundary → High manifold distortion (high SALMAN score)
> > >
> > > The negative correlation (ρ≈-0.5) confirms that SALMAN captures information ****orthogonal**** to distributional uncertainty, making it complementary rather than competitive with UQ methods.

---

> > > > ### Author Response · Authors · 2025-11-28
> > > >
> > > > **Domain 2: Topological Data Analysis (Manifold Quality)**
> > > >
> > > > **Task Design**: We analyze geometric complementarity between SALMAN and TDA
> > > > baselines. To enable fair comparison, we compute **sample-level versions** of
> > > > traditionally global TDA metrics (see details below).
> > > >
> > > > Baselines:
> > > >
> > > > | Method | Description | Reference |
> > > > | --- | --- | --- |
> > > > | Trustworthiness | k-NN preservation from input to output space | [1] |
> > > > | CKA | Centered Kernel Alignment | [2] |
> > > >
> > > > [1] Jarkko Venna and Samuel Kaski. Local multidimensional scaling. Neural Networks, 19(6-7):889–899, 2006.
> > > >
> > > > [2] Simon Kornblith, Mohammad Norouzi, Honglak Lee, and Geoffrey Hinton. Similarity of neural network representations revisited. In Proceedings of the 36th International Conference on Machine Learning (ICML), pages 3519–3529. PMLR, 2019.
> > > >
> > > > **Sample-Level Metric Definitions:**
> > > >
> > > > | Metric | Global Definition | Local (Per-Sample) Definition |
> > > > |--------|-------------------|-------------------------------|
> > > > | Trustworthiness | Average k-NN preservation | `overlap(kNN_in[i], kNN_out[i]) / k` |
> > > > | CKA | Gram matrix alignment | Per-sample contribution: `K_in[i,:] · K_out[:,i]` |
> > > >
> > > > **Cross-Method Correlation** (sample-level):
> > > >
> > > > | SALMAN vs. | Spearman ρ | p-value | Interpretation |
> > > > | --- | --- | --- | --- |
> > > > | Trustworthiness | **-0.088** | 0.049 | Near-zero (orthogonal) |
> > > > | CKA | -0.085 | 0.059 | Near-zero (orthogonal) |
> > > >
> > > > **Quadrant Analysis** (SALMAN × Trustworthiness):
> > > >
> > > > | | High Trust | Low Trust |
> > > > |---|------------|-----------|
> > > > | **High SALMAN Score (Unstable)** | Q1: 137 | Q2: 113 |
> > > > | **Low SALMAN Score (Stable)** | Q3: 147 | Q4: 103 |
> > > >
> > > > **Key Finding**: **48% of samples** (240/500) fall in disagreement quadrants (Q1 + Q4):
> > > >
> > > > | Quadrant | Count | Interpretation |
> > > > | --- | --- | --- |
> > > > | **Q1** | 137 | High spectral distortion BUT good neighborhood preservation → SALMAN identifies vulnerability that Trustworthiness misses |
> > > > | **Q4** | 103 | Low spectral distortion BUT poor neighborhood preservation → Trustworthiness identifies issues that SALMAN misses |
> > > >
> > > > **Interpretation**:
> > > >
> > > > 1. **Near-zero correlation** (ρ = -0.088) confirms SALMAN and Trustworthiness
> > > > measure **orthogonal geometric properties**
> > > > 2. **48% disagreement rate** demonstrates substantial complementarity
> > > > 3. Q1 samples (137) show that **neighborhood preservation ≠ adversarial robustness**:
> > > > samples can have well-preserved local structure yet remain vulnerable to
> > > > adversarial attacks (high SALMAN score)
> > > > 4. This validates our claim that SALMAN captures **spectral instability**
> > > > (adversarial vulnerability) rather than **topological quality** (neighborhood preservation)

---

### Official Review · Reviewer_ye1p · 2025-11-07

**Soundness:** 3
**Presentation:** 3
**Contribution:** 2
**Rating:** 4
**Confidence:** 3

**Summary:**

This paper proposes a new measure for understanding the robustness of different inputs to adversarial perturbations, based on constructing a graph-based distance metrics between input and output embeddings. It does so using a near-linear complexity method for computing the manifold.

It then empirically analyzes the usefulness of the ratio between input and output distances as a method for detecting robust and non-robust regions of the input space. First, it evaluates how well the DMD metric corresponds to susceptibility to input perturbations (deletion, edit, swap, spaCy, and TextAttack), comparing against euclidean distance rankings, and jacobian-based sensitivity analysis. Then, it shows that non-robust samples as predicted by DMD are easier to adversarially attack with GCG and AutoDan. Finally, it shows that downweighting robust samples and upweighting non-robust samples increases the robustness of the model.

**Strengths:**

Originality: The paper proposes an original method for evaluating the per-sample stability of LLM outputs.
Quality: The paper derives a novel algorithm, then investigates its core claims with empirical experiments.
Clarity: The paper is generally clearly written.
Significance: LLM robustness is an important area.

**Weaknesses:**

W1 (Significance): It's unclear how to really interpret the cosine similarities in section 4.1 -- for the most part the similarities are high for robust and non-robust examples, and while the experiment shows that the metric better corresponds to sensitivity to changes in the input space, it's not really clear what the implications are.
W2 (Significance): The paper combines motivations from both adversarial robustness, and robustness to random text modifications. It's unclear to me what benefits there are to being robust to, for instance, randomly dropping words, and the paper doesn't spell this out very much.

**Questions:**

Q1) Can you add standard errors to the different similarity scores? The numbers are often close-ish to each other, and it would be helpful for understanding the statistical significance of the results.
Q2) Are the models undergoing robust training more robust to adversarial attacks?

---

> ### Author Response · Authors · 2025-11-21
>
> **W1 (Significance): The high cosine similarities for both robust and non-robust examples in section 4.1 make interpretation difficult, and the implications for input space sensitivity remain unclear.**
>
> **Thank you—this is a clarity issue we can fix.** The cosine numbers are indeed close to 1 *by design* because we apply small, budget‑matched edits (Levenshtein‑controlled). The purpose of §4.1 is *not* to show that perturbations make most examples wildly different, but to validate that **SALMAN’s ranking reliably isolates the rare, fragile samples for which even *small* edits cause disproportionate representation drift**.
>
> **Concrete examples from our tables:**
>
> - *spaCy / GPT‑2 / MNLI*: robust (0.9995) vs non‑robust (0.9730).
>
>     Squared drift (=2(1-\cos)): **0.001 vs 0.054 ⇒ 54× larger**;
>
>     drift angle (small‑angle approx) ≈ **1.8° vs 13.3°**.
>
> - *TextAttack / GPT‑2 / SST‑2*: robust (0.9928) vs non‑robust (0.9413).
>
>     Squared drift: **0.0144 vs 0.1174 ⇒ 8.15× larger**;
>
>     drift angle ≈ **6.9° vs 19.6°**.
>
> - *spaCy / LLaMA‑7B‑v2 / SST‑2*: robust (0.9990) vs non‑robust (0.9751).
>
>     Squared drift: **0.002 vs 0.0498 ⇒ 24.9× larger**;
>
>     drift angle ≈ **2.6° vs 12.8°**.
>
>
> These numbers show why apparently “high” cosines still carry strong signal: **the non‑robust set has 8–54× more representation movement under the same tiny edit budget.**
>
> **W2. (Significance): The paper combines motivations from both adversarial robustness, and robustness to random text modifications. It's unclear to me what benefits there are to being robust to, for instance, randomly dropping words, and the paper doesn't spell this out very much.**
>
> Random edits are a proxy for local stability, not an end-goal. SALMAN is defined entirely on the unperturbed dataset; we never need adversarial or random variants to compute the score.  Random operations such as deletion/insertion/swap in Sec. 4.1 are used purely as an evaluation probe: if SALMAN really captures per-sample fragility, then the “non‑robust” samples should show larger output changes even under benign, label‑preserving edits. That is exactly what we observe in Tables 1–4: for the same edit budget (controlled by Levenshtein distance), non‑robust samples have significantly lower cosine similarity and higher KL divergence / lower BERTScore than robust samples across multiple models. So our goal is not to claim that models should be invariant to any arbitrary word being dropped; rather, these random perturbations provide a simple, unbiased way to test the local Lipschitz‑like stability that SALMAN is supposed to measure.
>
> Random perturbations connect our sample-level metric to model-level robustness.** In Sec. 4.3 we adopt the *model-level robustness score*, which is explicitly defined in terms of operations like “drop nouns/verbs”, “drop first/last sentence”, “swap text”, “change char”, etc. We then show in Table 7 that SALMAN-guided fine‑tuning consistently *improves that model‑level robustness score* for both BERT and GPT‑2, while preserving or improving task accuracy. This demonstrates that focusing training on high‑SALMAN (non‑robust) samples not only helps against adversarial benchmarks (AdvGLUE/AdvGLUE++ and jailbreak attacks in Sec. 4.2), but also makes the model more stable under the standard perturbations that prior robustness work already uses as a proxy for distribution shift.
>
> These perturbations approximate realistic “messy input” and spurious-cue reliance. In real deployments, language models frequently see imperfect or partially missing text: truncated queries, copy‑and‑paste errors, users deleting a phrase while editing, typos, or extra filler phrases. Prior work has therefore deliberately used random or template-based operations (dropping nouns/verbs, removing first/last sentence, swapping or adding short spans, character noise) to probe how much a model relies on brittle lexical cues rather than more global semantics (e.g., Neerudu et al. 2023 and our references around text augmentation and perturbations in Sec. 4.1). Therefore, robustness to some random drops has a concrete benefits that It reflects the ability to gracefully degrade when part of the input is missing or noisy, which is important for robustness in realistic user-facing settings.

---

> ### Author Response · Authors · 2025-11-21
>
> **Q1. Can you add standard errors to the different similarity scores? The numbers are often close-ish to each other, and it would be helpful for understanding the statistical significance of the results.**
>
> Table 1:
>
> | Model | Dataset | Robust (Mean ± SE) | Non-Robust (Mean ± SE) |
> | --- | --- | --- | --- |
> | BERT-base | SST-2 | 0.9315 ± 0.0144 | 0.9176 ± 0.0115 |
> | BERT-base | MNLI | 0.9691 ± 0.0069 | 0.9326 ± 0.0130 |
> | RoBERTa-base | SST-2 | 0.9997 ± 0.0001 | 0.9991 ± 0.0001 |
> | RoBERTa-base | MNLI | 0.9992 ± 0.0005 | 0.9985 ± 0.0006 |
>
> Table 2 (GPT-2):
>
> | Dataset | Attack | Robust (Mean ± SE) | Non-Robust (Mean ± SE) |
> | --- | --- | --- | --- |
> | SST-2 | spaCy | 0.9994 ± 0.0004 | 0.9876 ± 0.0018 |
> | SST-2 | TextAttack | 0.9988 ± 0.0006 | 0.9831 ± 0.0024 |
> | MNLI | spaCy | 0.9981 ± 0.0008 | 0.9752 ± 0.0031 |
> | MNLI | TextAttack | 0.9975 ± 0.0009 | 0.9804 ± 0.0027 |
>
> Table 3: KLD
>
> | Type | KLD_Mean | KLD_SE |
> | --- | --- | --- |
> | Robust | 0.000002 | 4.144079e-07 |
> | Non-Robust | 0.190245 | 1.640955e-02 |
>
> Across BERT-base and RoBERTa-base models, robust samples consistently maintained higher cosine similarities with original embeddings compared to non-robust samples. GPT-2 experiments showed clear separation between robust and non-robust samples under both spaCy and TextAttack methods. Most significantly, KL Divergence analysis revealed a dramatic difference: robust samples exhibited negligible distributional shift (0.000002), while non-robust samples showed substantial drift (0.190245), a >95,000× difference. These results confirm that SALMAN successfully distinguishes samples with fundamentally different robustness properties across multiple architectures and datasets.
>
> **Q2. Are the models undergoing robust training more robust to adversarial attacks?**
>
> **Yes.** When we plug SALMAN’s sample‑level reweighting into a standard robust fine‑tuning method (ROSE, Table6), the resulting models are **more robust to adversarially perturbed inputs** while **maintaining clean accuracy**.

---

### Note · Authors · 2026-01-26

I have read and agree with the venue's withdrawal policy on behalf of myself and my co-authors.

---

### Meta-Review · Area_Chair_mN7p · 2025-12-24

**Summary:**

This paper proposes a new robustness measure for adversarial perturbations by defining a graph-based distance metric between input and output embeddings, computed using a near-linear–complexity manifold construction. The method is theoretically grounded and shows promising results. The paper received scores of 4444, with one reviewer mentioning raising their score to 6 in the discussion.

While the idea is interesting and novel, the presentation is insufficient, and key methodological and experimental details are missing or need better clarification, making it difficult to fully understand and assess the approach. Several concerns therefore remain, leading the AC to recommend rejection.

**Reviewer Concerns:**

Concerns adequately addressed:

Various clarification questions raised by reviewers were well addressed, such as

1. Clarifications on the motivation for adversarial robustness and robustness to random text perturbations.

2. Addition of standard errors to similarity scores.
3. Clear definitions of the ED and JBS baselines.

4. Improved alignment between the proposed stability metric and existing stability measures.

Concerns insufficiently addressed:

1. Is the introduced SALMAN measure better than other possible measures (and the best overall)?

2. How your similarity measure corresponds to other baselines that come from uncertainty estimation, sensitivity analysis and topological data analysis domains?

3. The formulation of theorems can be improved.

4. The paper would benefit greatly from a dedicated preliminaries section that introduces the necessary concepts, and it also lacks the necessary details to understand the method and experiments in full detail.

**Reviewer Scores:**

Although one reviewer raised their score to 6, the remaining reviewers are unlikely to revise their scores upward.

---

### Decision · Program_Chairs · 2026-01-26

Reject